# Uncertainty and Scale-Calibrated Contrastive Federated Segmentation under Client Heterogeneity

## Abstract

Federated learning presents a promising approach for medical image segmentation, particularly in addressing data privacy concerns. However, it faces significant challenges due to data heterogeneity across participating clients. This heterogeneity introduces variations in data scales and distributions, making it difficult to balance spatial accuracy and feature similarity when managing multidimensional heterogeneous data. To address these challenges, we propose a novel **Uncertainty- and Scale-Calibrated Contrastive Federated Segmentation under Client Heterogeneity (SAFCF)** with two key approaches: (i) an **uncertainty-driven dynamic scale-adaptive weighted aggregation (DSWA)** method, which balances the influence of local client data scales and reduces model drift caused by data heterogeneity through the use of epistemic uncertainty in weighted aggregation, and (ii) a **contrastive federated segmentation loss (CFSL)**, a local loss function that effectively balances spatial accuracy and feature similarity at the pixel level of an image by combining modified Dice loss with improved contrastive loss. Additionally, epistemic uncertainty layer learns weight distributions to introduce uncertainty, further improving model robustness and enabling adaptive learning from diverse data during training. Our framework demonstrates substantial improvements on standard benchmark medical image segmentation datasets, especially under highly non-IID conditions, when compared to traditional algorithms.

## 1 Introduction

Medical data are typically confined to individual medical centers and hospitals Yang et al. (2019), which makes centralized aggregation impractical and potentially violates legal regulations. This is particularly challenging for segmentation tasks, where high-quality labeled data is scarce and often siloed across institutions. Federated learning (FL) addresses these issues by allowing distributed medical institutions to collaboratively train a shared model while keeping their data local and private Sheller et al. (2020); Dou et al. (2021). This approach has gained prominence in machine learning due to its ability to balance collaborative model development with stringent data privacy requirements McMahan et al. (2017); Karimireddy et al. (2020); Hu et al. (2022). FL is applied in various fields, including medical imaging Kaissis et al. (2020); Kumar et al. (2021), landmark classification Hsu et al. (2020), and object detection Liu et al. (2020b). While significant strides have been made in FL , it still confronts the persistent issue of client drift. This occurs due to imbalanced and non-independent identically distributed (non-IID) data distributions across clients. As a result, each client's local model tends to diverge from the optimal global objective Mendieta et al. (2022). During the local training phase of FL, each client's model optimizes towards its specific local minimum, potentially leading to overfitting on the local training data. This divergence in optimization objectives among clients can hinder the global model's ability to generalize well. When these disparate local models are aggregated, the global model may not align effectively with the varied local objectives Jiang et al. (2022), ultimately degrading the overall network performance Karimireddy et al. (2020); Li et al. (2022). This issue is partic-

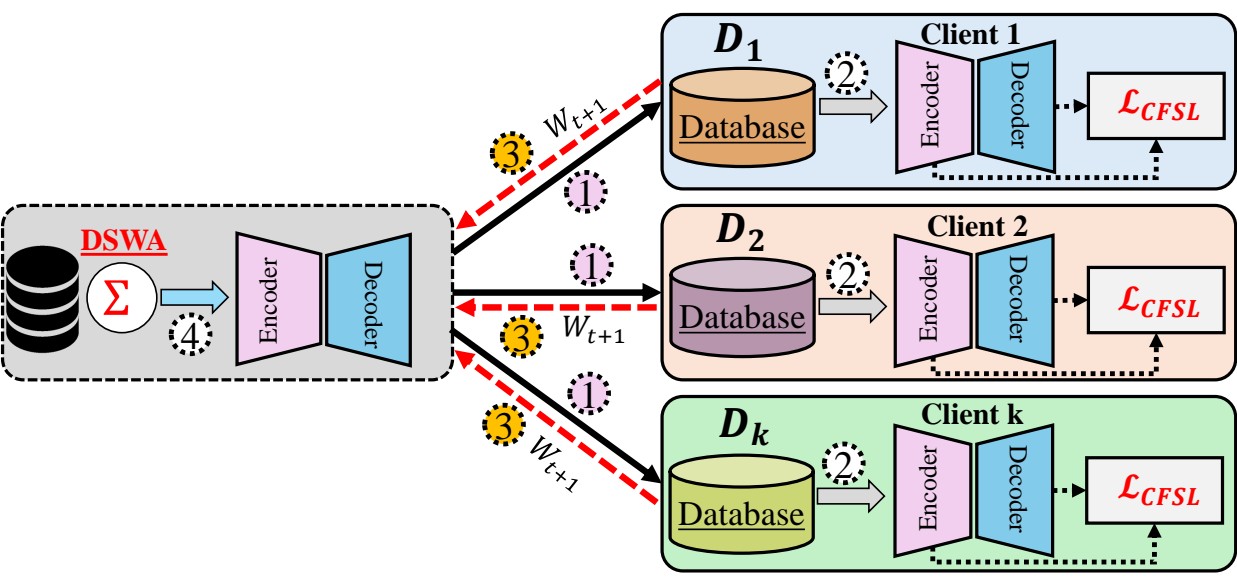

Figure 1: Overview of the proposed federated learning framework. This study primarily addresses the local training phase (step two) and the aggregation phase (step four)

ularly observed in medical applications, where data heterogeneity arising from regional, institutional, and equipment-related variations significantly impacts feature distributions.

Existing methods for addressing non-IID challenges include FedProx Li et al. (2020), which employs *L2-Norm* constraints on local updates, and SCAFFOLD Karimireddy et al. (2020), which applies variance reduction Johnson & Zhang (2013). For instance, Model-contrastive federated learning (MOON) incorporates model-level contrastive learning between the server and client models to align the global representation learned by clients Li et al. (2021). Methods such as FedMSRW Lu et al. (2024) focus on lesion segmentation using dynamic re-weighting, while FedLAW Liu et al. (2023) introduces learnable aggregation weights. However, these approaches do not explicitly incorporate scale-adaptive aggregation to address spatial and feature-level data heterogeneity.

Furthermore, these methods lack a mechanism to scale aggregation weights based on each client's dataset size ensuring that clients with more extensive or representative data contribute proportionately more to the global update. They do not incorporate uncertainty-driven techniques to mitigate model drift, a key factor in preventing overfitting to individual local datasets and enhancing the global model's generalizability. This is particularly relevant in real-world applications where clients possess datasets of varying sizes and variances. Thus, we propose the uncertainty-driven dynamic scale-adaptive weighted aggregation (DSWA) method to address these challenges. DSWA balances client contributions based on dataset scale and reduces model drift by incorporating epistemic uncertainty. Also, to improve the robustness and generalization of the segmentation model in handling data heterogeneity, we incorporate a variational dense layer that learns weight distributions, introducing epistemic uncertainty to capture variations across client data adaptively.

To address these non-IID challenges in federated learning, we propose the Uncertainty- and Scale-Calibrated Contrastive Federated Segmentation under Client Heterogeneity (SAFCF), which builds on novel methods in both the local training and aggregation phases (DSWA), as illustrated in Figure 8. The global model is a central aggregator that captures diverse patterns from various data distributions. To enhance knowledge transfer from the global model to local client models, we propose an improved contrastive loss function. This function enhances the similarity between current local and global model representations while simultaneously increasing the divergence between the current and previous local model representations, promoting effective

generalization across the network. Existing research Chen et al. (2020a); Oord et al. (2018) often struggles to balance spatial accuracy and feature similarity in multidimensional heterogeneous data. To address this challenge, we integrate the improved contrastive loss with a modified dice loss during local training to create the contrastive federated segmentation loss (CFSL). The CFSL effectively combines the precise localization capabilities of the modified Dice loss with the enhanced feature representation provided by the enhanced contrastive loss, resulting in more robust segmentation.

The primary contributions of our this work are: (i) We propose novel approaches in FL to address data heterogeneity, focusing on (a) the aggregation phase and (b) the local training phase. These approaches address the complexities posed by diverse datasets across different clients, (ii) Our DSWA method prioritizes clients based on data scale to ensure fair representation and mitigates model drift caused by data heterogeneity by incorporating epistemic uncertainty into the aggregation process, (iii) A novel CFSL is proposed to enhance local client learning by integrating spatial localization and model representation by combining a modified Dice loss with an enhanced contrastive loss and (iv) We introduce a holistic and effective training process for feature-based representation in segmentation models. The efficacy of our proposed SAFCF is demonstrated through experiments conducted on various medical image segmentation datasets.

## 2  Related Work

Recent research in federated contrastive learning addresses challenges with heterogeneous data, focusing on federated learning, its advancements in medical image segmentation, and contrastive learning.

**Federated Learning:** FedAvg McMahan et al. (2017) is an aggregation method in which the server averages model weights to form a global model for subsequent training rounds. Various enhancements have been proposed for FedAvg, particularly for non-IID data. FedProx Li et al. (2020) adds a proximal term to the local training objective, which constrains model updates by measuring the *L2-Norm* distance between local and global models. Scaffold Karimireddy et al. (2020) improves local updates through control variates and adjusting gradients based on differences between local and global control variates. However, these methods have primarily been evaluated on classification tasks rather than segmentation. FedMA Wang et al. (2020a) employs Bayesian non-parametric methods for layer-wise weight averaging, offering model flexibility but facing scalability issues with many clients. FedNova Wang et al. (2020b) improves communication efficiency by normalizing local updates, yet may struggle with extreme data heterogeneity. Diao et al. Diao et al. (2020) propose dynamic parameter allocation based on client capabilities, while Liang et al. Liang et al. (2020) simultaneously train local and global models, which can increase computational overhead. FedMix Yoon et al. (2021) combines data from multiple clients using a mixup operation, and FedFA Zhou & Konukoglu (2023) generates new training samples from a universal statistic to address feature drift. However, it adds complexity to maintaining consistent global statistics. Recent advances in federated learning for medical image segmentation have explored static weight aggregation and dynamic re-weighting based on task-specific loss functions.

While these methods address heterogeneity, federated medical image segmentation also requires strategies to tackle distribution shifts. Research focuses on challenges like distribution shift Jiang et al. (2023b); Wang et al. (2023), inconsistent ROIs Xu et al. (2023), and inter-client unfairness Jiang et al. (2023a). Approaches to enhance client training and reduce data drift include data pre-processing Sheller et al. (2020), domain-specific losses Gao et al. (2019), and contrastive learning Wu et al. (2021).

**Contrastive Learning:** Contrastive learning (CL), a self-supervised method for obtaining valuable visual representations from unlabeled data, has demonstrated state-of-the-art results in various approaches Chen et al. (2020b); He et al. (2020). The core idea of CL is to minimize the discrepancy between representations of augmented views of the same image (positive pairs) while maximizing the discrepancy between representations of views from different images (negative pairs). Notable CL frameworks are SimCLR Chen et al. (2020a) and InfoNCE Oord et al. (2018). SimCLR utilizes a similarity-based contrastive loss to generate positive pairs $(z_i, z_i^+)$ and negative pairs $(z_i, z_i^-)$ by employing two augmentations. These pairs are subse-

quently processed through an encoder to derive their corresponding embeddings. Li et al. Li et al. (2021) introduced MOON as a method to address the challenge of heterogeneous local data distributions among participants. Recent work, relaxed contrastive learning (RCL) enhances federated learning by mitigating gradient inconsistency across clients and preventing representation collapse Seo et al. (2024). However, these works primarily targets image-level classification tasks, making it less suited for segmentation tasks where pixel-level spatial and semantic details are essential. As a result, the problem of client drift in federated learning (FL) networks for segmentation remains an open area for investigation.

Motivated by this, we developed an enhanced contrastive loss function for effective segmentation that captures intricate patterns in feature representations across multi-dimensions of an image, distinguishing it from traditional methods.

**Uncertainty Estimation:** Uncertainty estimation assigns confidence levels to a model's output. Approximate Bayesian methods and deterministic uncertainty methods Liu et al. (2020a); Postels et al. (2021) have shown strong performance in quantifying epistemic uncertainty, especially in computer vision tasks, making them more applicable to scenarios like semantic segmentation. Monte-Carlo Dropout Gal & Ghahramani (2016) is easy to implement but can be unreliable, while Deep Ensembles Lakshminarayanan et al. (2017) are effective but computationally expensive.

In this work, we focus on the supervised learning setting in which we simultaneously address local (CFSL) and global (DSWA) strategies in our federated framework (SAFCF) to effectively handle heterogeneous data sources on individual clients for robust segmentation.

## 3 SAFCF: Methodology

Our (SAFCF) (Figure 2) addresses non-IID data by learning a global model on the server that aggregates client updates and captures diverse cross-client representations. Each input first passes through a variational dense layer that models epistemic uncertainty by learning a distribution over weights; the sampled parameters are then used in the encoder–decoder to improve robustness under data variability. During local training, we extract feature representations from the encoder's final convolutional layer and optimize an advanced contrastive objective to align the current client representation with the global representation while discouraging similarity to the client's previous-state representation. Concretely, let $f_\theta(\cdot)$ denote the encoder feature mapping with parameters $\theta$. For an input $x$, we define $\mathbf{h}_c = f_{\theta_i^t}(x)$ (current client model), $\mathbf{h}_s = f_{\theta^t}(x)$ (global/server model), and $\mathbf{h}_p = f_{\theta_i^{t-1}}(x)$ (previous client model). The contrastive loss increases similarity between $\mathbf{h}_c$ and $\mathbf{h}_s$ and decreases similarity between $\mathbf{h}_c$ and $\mathbf{h}_p$.

### 3.1 Contrastive Federated Segmentation Loss (CFSL)

To improve robustness under non-IID client data, we propose the **Contrastive Federated Segmentation Loss (CFSL)**, which jointly optimizes (i) *voxel-level localization* via a modified Dice formulation and (ii) *representation transfer* via an enhanced contrastive objective. CFSL is applied at each client during local training and is defined as

$$\mathbb{L}_{\text{CFSL}} = \lambda \mathbb{L}_c + (1-\lambda)\mathbb{L}_{\text{dice}}, \qquad \lambda \in [0,1], \tag{1}$$

where $\lambda$ balances representation alignment ($\mathbb{L}_c$) and spatial precision ($\mathbb{L}_{\text{dice}}$).

**Representations.** For an input volume $x$ at communication round $t$, we extract encoder representations at the final convolutional layer from: (i) the current local model $\Upsilon_i^t$, (ii) the previous local model $\Upsilon_i^{t-1}$, and (iii) the global server model $\Upsilon^t$, denoted as

$$\mathbf{h}_c = V_{\Upsilon_i^t}(x), \quad \mathbf{h}_p = V_{\Upsilon_i^{t-1}}(x), \quad \mathbf{h}_s = V_{\Upsilon^t}(x), \tag{2}$$

where $V_\Upsilon(\cdot)$ denotes the representation mapping induced by model $\Upsilon$. (All element-wise operations below are applied channel-wise and broadcast over spatial/voxel indices when $\mathbf{h}$ is a tensor.)

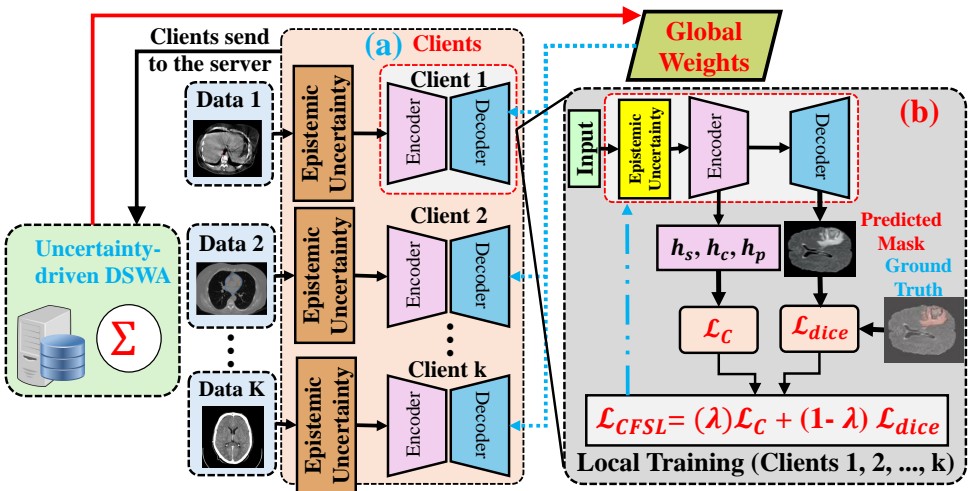

Figure 2: Overview of SAFCF: (a) shows the system with multiple clients, and (b) illustrates local training at each client. Our novel approaches include (i) CFSL, which balances spatial precision and feature similarity using $L_{CFSL}$, and (ii) DSWA, which adjusts aggregation for varying local data scales and mitigates model drift with epistemic uncertainty. The variational dense layer learns weight distributions to introduce uncertainty. Here, $h_c$ is from the current local model, $h_p$ from the previous local model, and $h_s$ from the global model.

**Enhanced contrastive loss** $\mathbb{L}_c$. We first compute pairwise similarities using temperature-scaled cosine similarity. Given two representations $\mathbf{a}, \mathbf{b} \in \mathbb{R}^k$, we define

$$\phi(\mathbf{a}, \mathbf{b}) = \frac{1}{\tau} \cdot \frac{\mathbf{a}^\top \mathbf{b}}{\|\mathbf{a}\|_2 \|\mathbf{b}\|_2 + \epsilon}, \tag{3}$$

where $\tau > 0$ is the temperature coefficient controlling similarity sharpness and $\epsilon$ ensures numerical stability. Using equation 3, we define

$$\phi_s = \phi(\mathbf{h}_s, \mathbf{h}_c), \qquad \phi_c = \phi(\mathbf{h}_c, \mathbf{h}_p). \tag{4}$$

Here, $\phi_s$ measures *current-local to global* alignment, while $\phi_c$ measures *current-local to previous-local* similarity. We then compute a relative alignment score

$$\beta = \phi_s - \phi_c, \tag{5}$$

which acts as a margin-like signal: a larger $\beta$ indicates that the current local representation is more consistent with the global representation than with the previous local state, strengthening knowledge transfer while discouraging temporal collapse.

Next, we compute a distance term that quantifies how far the current representation deviates from this relative margin:

$$\Delta = \max\left(\sqrt{\zeta_s\big((\mathbf{h}_c^2 - \beta^2)^2\big)}, \epsilon\right), \tag{6}$$

where $\zeta_s(\cdot)$ denotes a reduced-sum operator over spatial/feature dimensions, and $\epsilon$ is a small constant ensuring numerical stability and non-degenerate gradients (i.e., preventing the distance from collapsing to 0).

We then form an intermediate contrastive quantity $\eta$ that increases divergence from the previous local state while maintaining alignment structure:

$$\eta = \big(\mathbf{h}_p \odot \mathbf{h}_c^2\big) + \big((1 - \mathbf{h}_p) \odot \Delta\big), \tag{7}$$

where $\odot$ is the Hadamard product. *Intuition:* $\mathbf{h}_p$ acts as a soft weighting of the previous-state influence—where $\mathbf{h}_p$ is large, the term $\mathbf{h}_p \odot \mathbf{h}_c^2$ emphasizes consistency with the current features; where $\mathbf{h}_p$ is small, $(1 - \mathbf{h}_p) \odot \Delta$ increases the separation pressure via $\Delta$, discouraging the current representation from collapsing into the previous local representation under heterogeneity.

To keep the scaling of the objective consistent across clients and rounds, we compute

$$\chi = \sqrt{\zeta_s(\mathbf{h}_s^2)}, \tag{8}$$

which down-scales the server representation magnitude into a comparable range, avoiding overly large contrastive signals when $\mathbf{h}_s$ has high energy.

Finally, the enhanced contrastive loss is defined as

$$\mathbb{L}_c = \frac{\zeta_m(\eta) \ - \ \zeta_m(\eta) \cdot \alpha}{\log(\chi)} \ = \ \frac{(1 - \alpha)\,\zeta_m(\eta)}{\log(\chi)}, \tag{9}$$

where $\zeta_m(\cdot)$ denotes a reduced-mean operator. *Intuition:* the numerator scales the strength of the contrastive penalty via $(1-\alpha)$, while the $\log(\chi)$ term provides a magnitude-aware normalization so that updates do not become overly aggressive when the server feature energy increases.

**Batch-wise adaptive scaling of $\alpha$ via triplet-mean ratio (tmr).** We update the scaling factor $\alpha$ using a triplet-mean ratio computed from the three representations:

$$\varpi_{ptmr} = \zeta_m(\mathbf{h}_p), \quad \varpi_{ctmr} = \zeta_m(\mathbf{h}_c), \quad \varpi_{stmr} = \zeta_m(\mathbf{h}_s), \tag{10}$$

$$\approx\!\!\succ\!\!\searrow \ = \frac{\zeta_m(\varpi_{stmr}, \varpi_{ctmr}, \varpi_{ptmr})}{\varpi_{stmr} + \varpi_{ctmr} + \varpi_{ptmr}}, \tag{11}$$

and the batch-wise update is

$$\alpha = \approx\!\!\succ\!\!\searrow \cdot \alpha_p, \tag{12}$$

where $\alpha_p$ is the previous value of $\alpha$. *Intuition:* $\approx\!\!\succ\!\!\searrow$ summarizes the relative scale of the current/global/previous representations in the batch and uses it to adapt the contrastive scaling, helping keep optimization stable across heterogeneous feature magnitudes and across communication rounds.

**Modified Dice loss $\mathbb{L}_{\mathbf{dice}}$ (3D voxel-wise).** For segmentation supervision, we use the following modified Dice formulation over 3D voxel indices $(i, j, k)$:

$$\mathbb{L}_{\text{dice}} = \frac{2 \sum_{i,j,k} p_{\text{true}_{ijk}} \, p_{\text{pred}_{ijk}}}{\sum_{i,j,k} p_{\text{true}_{ijk}}^2 + \sum_{i,j,k} p_{\text{pred}_{ijk}}^2 + \epsilon}, \tag{13}$$

where $p_{\text{true}_{ijk}}$ and $p_{\text{pred}_{ijk}}$ denote the ground-truth and predicted voxel values, respectively, and $\epsilon$ ensures numerical stability.

**Overall intuition.** CFSL explicitly couples *spatial supervision* (modified Dice for accurate 3D localization) with *representation-level guidance* (enhanced contrastive loss) so that each client learns locally useful segmentation while remaining aligned with the global representation under strong inter-client distribution shifts.

## 3.2 Uncertainty-driven Dynamic Scale-adaptive Weighted Aggregation (DSWA)

To mitigate client drift under non-IID data, we propose **DSWA**, a server-side aggregation rule that (i) balances client scale and (ii) down-weights uncertain client updates. Let $\mathbf{w}_i^t$ denote the model parameters

returned by client $i$ at communication round $t$, and let $n_i = |D_i|$ be its local sample size. We first compute the client scale ratio

$$\gamma_i = \frac{n_i}{\sum_{j=1}^{k} n_j}, \tag{14}$$

and use its complementary form to prevent large clients from dominating,

$$\tilde{\gamma}_i = \frac{1 - \gamma_i}{\sum_{j=1}^{k}(1 - \gamma_j)}. \tag{15}$$

**Second-moment uncertainty (proxy).** We estimate client uncertainty via the empirical second central moment of parameters around the (scale-balanced) mean:

$$\bar{\mathbf{w}}^t = \sum_{j=1}^{k} \tilde{\gamma}_j \, \mathbf{w}_j^t, \tag{16}$$

$$\mathbf{u}_i^t = (\mathbf{w}_i^t - \bar{\mathbf{w}}^t)^{\odot 2}, \qquad U_i^t = \zeta_m(\mathbf{u}_i^t), \tag{17}$$

where $\odot$ denotes the Hadamard product and $\zeta_m(\cdot)$ is a reduced-mean operator (yielding a scalar second-moment estimate). Intuitively, larger $U_i^t$ indicates higher disagreement from the global consensus, and is treated as higher epistemic uncertainty (proxy).

**Dynamic scale-adaptive weights.** We convert uncertainty into a reliability score and combine it with scale balancing:

$$\omega_i^t = \frac{1}{U_i^t + \epsilon}, \qquad a_i^t = \frac{\tilde{\gamma}_i \, \omega_i^t}{\sum_{j=1}^{k} \tilde{\gamma}_j \, \omega_j^t}, \tag{18}$$

and update the server model as

$$\mathbf{w}_s^{t+1} = \sum_{i=1}^{k} a_i^t \, \mathbf{w}_i^t. \tag{19}$$

This update is recomputed each round from the submitted client models (which themselves are learned over mini-batches), yielding a dynamic, drift-aware aggregation under heterogeneous client data.

## 4  Experiments

**Dataset Details: UPenn-GBM Dataset**: This dataset Bakas et al. (2022) includes multi-parametric MRI scans from 630 patients with Glioblastoma (GBM), featuring patient demographics, clinical outcomes, genomic data, and tumor segmentations. **UCSF PDGM Dataset**: The UCSF dataset Calabrese et al. (2022) contains preoperative MRI scans from 500 subjects with diffuse gliomas, including diffusion and perfusion MRI, tumor segmentations, and genetic and survival data. **MSD Dataset**: The MSD dataset Antonelli et al. (2022) includes three subsets. The **Brain** dataset contains 750 mp-MRI images from glioma patients, derived from the 2016 and 2017 challenges. The **Heart** dataset consists of 30 MRI scans of the left atrium, known for high anatomical variability, from the 2013 LASC. The **Hippocampus** dataset comprises 195 MRI images focused on hippocampal segmentation, collected at Vanderbilt University Medical Center.

We use a 70:20:10 train-validation-test split for each dataset. For the MRI brain images, we preprocess the data by scaling and standardizing all images to a uniform size of $128 \times 128 \times 128 \times 1$, where $128 \times 128 \times 128$ denotes the image dimensions and the last dimension represents the channels.

**Implementation Details:** To implement the SAFCF setup and the other baseline methods, we employ the TensorFlow framework. We used a batch size of 8 for our experimental setup across the training of various datasets. We utilize Adam optimizer with optimal learning rate $\alpha$ set to $e - 5$. Adam's first momentum is set to 0.9, and the second is configured to 0.999. The number of local epochs is 40 for 10, 50, and 100

---
**Algorithm 1** SAFCF Training with CFSL (client) and DSWA (server)
---
 1: **Input:** communication rounds $T$, clients $K$, local epochs $E$, learning rate $\alpha$, CFSL weight $\lambda$, stability $\epsilon$
 2: **Output:** final global model parameters $\mathbf{w}_s^T$
 3: **Server:** Initialize global parameters $\mathbf{w}_s^0$
 4: Obtain client sizes $n_i \leftarrow |D_i|$ for all $i \in \{1, \ldots, K\}$
 5: **for** $t = 0, 1, \ldots, T - 1$ **do**
 6:     **for all** clients $i = 1, \ldots, K$ **in parallel do**
 7:         Send $\mathbf{w}_s^t$ to client $i$
 8:         $\mathbf{w}_i^t \leftarrow \text{CLIENTTRAINING}(i, \mathbf{w}_s^t)$
 9:     **end for**
  **DSWA (server aggregation):**
10:     $\mathbf{w}_s^{t+1} \leftarrow \text{DSWA}(\{\mathbf{w}_i^t\}_{i=1}^K, \{n_i\}_{i=1}^K, \epsilon)$
                                  $\triangleright$ See Eqs. equation 14–equation 19 for $\gamma_i, \tilde{\gamma}_i, U_i^t, \omega_i^t, a_i^t$ and update rule.
11: **end for**
12: **return** $\mathbf{w}_s^T$
    **ClientTraining**$(i, \mathbf{w}_s^t)$:
13: $\mathbf{w} \leftarrow \mathbf{w}_s^t$
14: **for** epoch $e = 1, \ldots, E$ **do**
15:     **for** each batch $b = \{x, y\}$ from $D_i$ **do**
16:         Compute $\mathbb{L}_c$ using $(\mathbf{h}_c, \mathbf{h}_p, \mathbf{h}_s)$ as defined in Sec. 3.1
17:         Compute modified Dice loss $\mathbb{L}_{\text{dice}}$
18:         $\mathbb{L}_{\text{CFSL}} \leftarrow \lambda \mathbb{L}_c + (1 - \lambda) \mathbb{L}_{\text{dice}}$
19:         $\mathbf{w} \leftarrow \mathbf{w} - \alpha \nabla_{\mathbf{w}} \mathbb{L}_{\text{CFSL}}$
20:     **end for**
21: **end for**
22: **return w**
---

communication rounds at each client across the overall FL approach. We utilized the Nvidia V100 GPU to evaluate all other baselines within our federated framework.

**Result Analysis:** To demonstrate the effectiveness of our loss function $\mathbb{L}_{CFSL}$, we conducted a comparative analysis with several baseline methods, including SimCLR Chen et al. (2020a), InfoNCE Oord et al. (2018), Vanilla Dice Sudre et al. (2017), LocalCL Chaitanya et al. (2020), SwAV Caron et al. (2020) and Focal Tversky loss Abraham & Khan (2019) in Table 1. We implement these approaches with our uncertainty-driven DSWA aggregation method in our federated framework, so we named these federated variants FedDice, FedInfoNCE, FedSimCLR, FedFoc-Tvers, FedLocal-CL, and FedSwAV. Our experiments utilize three distinct datasets across different clients: UCSF PDGB (client-1), UPenn-GBM (client-2), and MSD (client-3). By employing different datasets, we address challenges related to varying data scales and distributions. A 3D U-Net architecture is employed, with a variational dense layer as the initial layer, which processes inputs through epistemic uncertainty to enhance representation learning for effective image segmentation across all clients. Dice coefficient (Dice), precision (Prec.), sensitivity (Sens.), and IoU are used as metrics for evaluation. At CR = 100, our SAFCF model outperformed several baselines across all datasets. For the UCSF-PDGM dataset, SAFCF surpassed FedDice by 5.6%, FedInfoNCE by 13.6%, FedSimCLR by 9.0%, FedLocalCL by 11.4%, FedSwAV by 13.6%, and FedFocalTversky by 4.5% in Dice coefficient. On the UPenn-GBM dataset, SAFCF exceeded FedDice by 8.6%, FedInfoNCE by 13.0%, FedSimCLR by 9.8%, FedLocalCL by 11.9%, FedSwAV by 5.4%, and FedFocalTversky by 7.6%. For the MSD dataset, SAFCF demonstrated superior performance over FedDice by 8.4%, FedInfoNCE by 8.6%, FedSimCLR by 10.5%, FedLocalCL by 12.6%, FedSwAV by 18.9%, and FedFocalTversky by 23.2%. SAFCF is comparable to the other baselines for the first ten communication round. However, CFSL exhibits better and faster performance improvement for the 50 and 100 communication rounds. This indicates its ability to capture intricate patterns in feature

| Data | Baselines | CR = 10 | | | | CR = 50 | | | | CR = 100 | | | |
|---|---|---|---|---|---|---|---|---|---|---|---|---|---|
| | | Dice ↑ | IoU ↑ | Prec. ↑ | Sens. ↑ | Dice ↑ | IoU ↑ | Prec. ↑ | Sens. ↑ | Dice ↑ | IoU ↑ | Prec. ↑ | Sens. ↑ |
| **UCSF PDGM** | FedDice | 0.70 | 0.55 | **0.77** | 0.65 | 0.77 | 0.64 | 0.75 | 0.79 | 0.83 | 0.71 | 0.79 | 0.85 |
| | FedFoc-Tvers | 0.68 | 0.51 | 0.55 | 0.78 | 0.74 | 0.64 | 0.71 | 0.77 | 0.84 | 0.64 | 0.82 | 0.88 |
| | FedInfoNCE | 0.63 | 0.46 | 0.57 | 0.69 | 0.70 | 0.57 | 0.64 | 0.73 | 0.76 | 0.66 | 0.73 | 0.78 |
| | FedSimCLR | 0.68 | 0.48 | 0.65 | 0.70 | 0.75 | 0.63 | 0.73 | 0.78 | 0.80 | 0.68 | 0.79 | 0.83 |
| | FedLocalCL | 0.66 | 0.49 | 0.66 | 0.69 | 0.72 | 0.61 | 0.69 | 0.75 | 0.78 | 0.67 | 0.76 | 0.79 |
| | FedSwAV | 0.70 | 0.51 | 0.63 | 0.72 | 0.71 | 0.62 | 0.68 | 0.74 | 0.76 | 0.64 | 0.75 | 0.78 |
| | **SAFCF** | **0.75** | **0.60** | **0.77** | **0.79** | **0.82** | **0.69** | **0.79** | **0.83** | **0.88** | **0.79** | **0.85** | **0.91** |
| **UPenn-GBM** | FedDice | 0.73 | 0.60 | 0.74 | 0.75 | 0.79 | 0.65 | 0.74 | 0.81 | 0.84 | 0.73 | 0.81 | 0.87 |
| | FedFoc-Tvers | 0.75 | 0.60 | 0.65 | 0.79 | 0.81 | 0.69 | 0.78 | 0.81 | 0.85 | 0.74 | 0.82 | 0.87 |
| | FedInfoNCE | 0.69 | 0.53 | 0.74 | 0.64 | 0.76 | 0.62 | 0.72 | 0.78 | 0.80 | 0.69 | 0.79 | 0.82 |
| | FedSimCLR | 0.71 | 0.58 | 0.75 | 0.69 | 0.77 | 0.64 | 0.74 | 0.79 | 0.83 | 0.72 | 0.80 | 0.85 |
| | FedLocalCL | 0.69 | 0.56 | 0.72 | 0.70 | 0.75 | 0.61 | 0.71 | 0.78 | 0.81 | 0.70 | 0.79 | 0.83 |
| | FedSwAV | 0.76 | 0.61 | 0.66 | **0.81** | 0.81 | 0.68 | 0.77 | 0.82 | 0.87 | 0.75 | 0.81 | 0.90 |
| | **SAFCF** | **0.78** | **0.64** | **0.82** | **0.81** | **0.85** | **0.73** | **0.81** | **0.87** | **0.92** | **0.81** | **0.85** | **0.94** |
| **MSD (Brain)** | FedDice | 0.77 | 0.63 | 0.77 | 0.79 | 0.82 | 0.71 | 0.80 | 0.83 | 0.87 | 0.75 | 0.85 | 0.89 |
| | FedFoc-Tvers | 0.65 | 0.49 | 0.50 | 0.75 | 0.70 | 0.61 | 0.68 | 0.72 | 0.73 | 0.61 | 0.70 | 0.76 |
| | FedInfoNCE | 0.75 | 0.71 | 0.80 | 0.85 | 0.80 | 0.69 | 0.76 | 0.82 | 0.83 | 0.74 | 0.81 | 0.85 |
| | FedSimCLR | 0.75 | 0.73 | 0.81 | 0.86 | 0.81 | 0.72 | 0.79 | 0.83 | 0.85 | 0.74 | 0.86 | 0.89 |
| | FedLocalCL | 0.74 | 0.71 | 0.80 | 0.85 | 0.79 | 0.69 | 0.78 | 0.81 | 0.83 | 0.71 | 0.80 | 0.86 |
| | FedSwAV | 0.68 | 0.51 | 0.55 | 0.78 | 0.72 | 0.65 | 0.71 | 0.75 | 0.77 | 0.68 | 0.76 | 0.79 |
| | **SAFCF** | **0.86** | **0.72** | **0.88** | **0.89** | **0.91** | **0.79** | **0.89** | **0.94** | **0.95** | **0.86** | **0.91** | **0.96** |

Table 1: Comparison of our proposed approach (SAFCF) with contrastive and segmentation loss baselines. Results are presented for 10, 50, and 100 communication rounds (CR) with 3-fold cross-validation across the UCSF PDGM, UPenn-GBM, and MSD (Brain) datasets at each individual client.

representations across all three dimensions of an image due to extracting spatial and feature represe at the model level. CFSL can handle the local update drifts throughout the training process, helping the model $\Upsilon_i$ to converge faster.

| Agg Method | Dice ↑ | IoU ↑ | Prec. ↑ | Sens. ↑ |
|---|---|---|---|---|
| FedAvg | 0.51±0.01 | 0.35±0.01 | 0.66±0.02 | 0.57±0.03 |
| FedProx | 0.70±0.03 | 0.62±0.01 | 0.70±0.02 | 0.86±0.03 |
| Scaffold | 0.76±0.01 | 0.60±0.01 | 0.78±0.03 | 0.76±0.01 |
| FedNova | 0.74±0.02 | 0.61±0.03 | 0.72±0.01 | 0.76±0.30 |
| FedMA | 0.75±0.01 | 0.61±0.03 | 0.73±0.01 | 0.77±0.02 |
| **DSWA** | **0.79±0.02** | **0.64±0.03** | **0.78±0.01** | **0.81±0.01** |

Table 2: Comparison of various aggregation methods and their impact on server performance.

**Additional Empirical Analysis and Ablation Studies:** DSWA is evaluated on merged client's test data and compared against server aggregation methods such as FedAvg, FedProx, and Scaffold and observed that our DSWA outperforms other methods, as shown in Table 2. The DSWA outperforms FedAvg, Fedprox, Scaffold, FedNova, and FedMA by the margin of 35.4%, 11.4%, 3.8%, 6.3%, and 5.1% respectively. It enhances segmentation precision and consistency by balancing local data scales and mitigating model drift through epistemic uncertainty in the weighted aggregation process. This framework enhances both convergence speed and segmentation accuracy compared to state-of-the-art models, effectively managing heterogeneous data from diverse brain tumor sources.

Table 3 presents a performance comparison between the proposed SAFCF framework and state-of-the-art (SOTA) methods such as MOON, FedMix, and FedRCL on UCSF PDGM, UPenn-GBM, and MSD datasets at each client. SAFCF outperforms these methods with higher robustness and accuracy in handling heterogeneous medical image segmentation. MOON's model-level contrastive learning is effective for classification but lacks pixel-level precision, limiting its segmentation performance. FedMix's mixup approach blurs spatial details, reducing structural accuracy, while FedRCL, though enhancing feature diversity, lacks scale-adaptive aggregation and precise pixel alignment. SAFCF overcomes these issues by combining scale-adaptive aggre-

| Data | UCSF PDGM | UPENN-GBM | MSD |
|------|-----------|-----------|-----|
| MOON | 0.74 ± 0.01 | 0.76 ± 0.02 | 0.77 ± 0.03 |
| FedMix | 0.79 ± 0.02 | 0.80 ± 0.04 | 0.81 ± 0.02 |
| FedRCL | 0.71 ± 0.02 | 0.72 ± 0.04 | 0.78 ± 0.02 |
| **SAFCF** | **0.88 ± 0.03** | **0.92 ± 0.03** | **0.95 ± 0.01** |

Table 3: Comparison of proposed framework performance with SOTA methods at each client.

gation with contrastive segmentation loss, achieving superior results across varied data scales and improving spatial feature alignment crucial for medical image segmentation.

| Approach | Data | Dice ↑ | IoU ↑ | Prec. ↑ | Sens. ↑ |
|----------|------|--------|-------|---------|---------|
| DSWA | UCSF PDGM | 0.68±0.03 | 0.48±0.06 | 0.65±0.02 | 0.70±0.02 |
| | UPenn-GBM | 0.71±0.01 | 0.58±0.03 | 0.75±0.03 | 0.69±0.02 |
| | MSD | 0.75±0.04 | 0.73±0.02 | 0.81±0.04 | 0.86±0.03 |
| CFSL | UCSF PDGM | 0.57±0.04 | 0.48±0.04 | 0.62±0.02 | 0.68±0.01 |
| | UPenn-GBM | 0.53±0.02 | 0.41±0.02 | 0.59±0.03 | 0.52±0.02 |
| | MSD | 0.56±0.03 | 0.51±0.03 | 0.62±0.02 | 0.62±0.03 |
| **SAFCF** | UCSF PDGM | **0.88±0.03** | **0.79±0.04** | **0.85±0.04** | **0.91±0.04** |
| | UPenn-GBM | **0.92±0.03** | **0.81±0.03** | **0.85±0.01** | **0.94±0.02** |
| | MSD | **0.95±0.01** | **0.86±0.05** | **0.91±0.02** | **0.96±0.03** |

Table 4: Ablation results for the different components of SAFCF.

**Effectiveness of Each Component:** To thoroughly examine the impact and contributions of SAFCF's core methodologies, DSWA and CFSL, we conducted a comprehensive analysis over 100 communication rounds, as detailed in Table 4. Initially, using DSWA with basic dice segmentation loss, we observed dice coefficients of 68% for UCSF PDGM, 71% for UPenn-GBM, and 75% for MSD. When CFSL was used alone with FedAvg, the dice coefficients dropped to 57% for UCSF PDGM, 53% for UPenn-GBM, and 56% for MSD. Finally, within the SAFCF framework, incorporating both CFSL and DSWA, achieved significantly improved dice coefficients of 88% for UCSF PDGM, 92% for UPenn-GBM, and 95% for MSD. These results highlight CFSL and DSWA's key roles in improving tumor segmentation through effective feature alignment and uncertainty-driven aggregation across diverse data sources.

**Effectiveness on Distinct Medical Datasets**: Our method was evaluated on distinct anatomical datasets from MSD dataset at each client (Table 5) over 100 communication rounds. The highest performance was observed with the Hippocampus dataset (Client-1) followed by the Brain Tumor dataset (Client-3), and the Heart dataset (Client-2). These results demonstrate the SAFCF effectiveness across various organs, maintaining high precision and sensitivity across all datasets.

**Scalability:** To test the scalability of our SAFCF, we performed experiments involving 5 to 80 clients, as detailed in Table 6, revealed that increasing the number of clients led to enhanced model performance, as evidenced by improvements in the Dice coefficient, IoU, Precision, and sensitivity. Data from UCSF PDGM, UPenn-GBM and MSD datasets have been distributed randomly among the clients. However, this performance gain came at the cost of increased communication and computation, resulting in extended training times.

## 5 More Analysis

We conducted extensive empirical analysis and ablation studies of our SAFCF framework, focusing on its generalizability, performance in a homogeneous setup, ability to handle limited domain shifts, scalability, effectiveness with various segmentation models, hyper-parameter tuning, communication efficiency, and qualitative results. To ensure reproducibility, all experiments were performed with a consistent random seed of 42.

| Data | Dice ↑ | IoU ↑ | Prec. ↑ | Sens. ↑ |
|---|---|---|---|---|
| Hippocampus | 0.98 ± 0.01 | 0.87 ± 0.02 | 0.95 ± 0.03 | 0.97 ± 0.01 |
| Heart | 0.88 ± 0.02 | 0.76 ± 0.04 | 0.85 ± 0.02 | 0.90 ± 0.01 |
| Brain | 0.92 ± 0.01 | 0.83 ± 0.02 | 0.88 ± 0.01 | 0.93 ± 0.03 |

Table 5: Performance of the proposed approach across various MSD datasets at individual clients.

| Number of Clients | Dice ↑ | IoU ↑ | Prec. ↑ | Sens. ↑ |
|---|---|---|---|---|
| 5 | 0.89 ± 0.02 | 0.80 ± 0.03 | 0.87 ± 0.01 | 0.92 ± 0.03 |
| 10 | 0.91 ± 0.01 | 0.82 ± 0.02 | 0.88 ± 0.03 | 0.93 ± 0.02 |
| 30 | 0.92 ± 0.02 | 0.84 ± 0.01 | 0.89 ± 0.04 | 0.95 ± 0.01 |
| 50 | 0.94 ± 0.01 | 0.87 ± 0.03 | 0.91 ± 0.02 | 0.96 ± 0.03 |
| 80 | 0.93 ± 0.02 | 0.85 ± 0.02 | 0.91 ± 0.01 | 0.95 ± 0.02 |

Table 6: Impact of the number of clients on model performance.

## 5.1 Generalizability

The results in Table 7 demonstrate the performance of our proposed SAFCF framework on three breast ultrasound datasets, each placed at a different client and evaluated using the 2D U-Net segmentation model. The experimental setup follows the methodology outlined in FedMix Wicaksana et al. (2022). The framework consistently performs well across these datasets, with slight variations in the performance metrics reflecting the inherent variability of the data. The high dice coefficients, along with consistent precision and sensitivity scores, suggest that the framework is robust and adaptable to different medical image segmentation datasets.

| Data | Dice ↑ | IoU ↑ | Precision ↑ | Sensitivity ↑ |
|---|---|---|---|---|
| BUSIS | 0.89 ± 0.01 | 0.78 ± 0.02 | 0.90 ± 0.01 | 0.91 ± 0.03 |
| BUS | 0.86 ± 0.02 | 0.75 ± 0.03 | 0.89 ± 0.01 | 0.88 ± 0.01 |
| UDIAT | 0.87 ± 0.03 | 0.76 ± 0.01 | 0.88 ± 0.02 | 0.89 ± 0.03 |

Table 7: Proposed Approach on Breast Ultrasound Datasets each placed at three individual clients with 2D U-Net as Segmentation Model.

## 5.2 Homogeneous Setup

We conducted an experiment using independent and identically distributed (IID) data to evaluate the impact of uniform data distribution on our SAFCF framework. To ensure uniformity, the same dataset was used across all clients, with the data evenly distributed so that each client received an equal number of samples (Table 8). For example, in the MSD dataset, each client was allocated 250 samples, with 190 for training and the remainder for validation and testing. The UCSF-PDGM and UPenn-GBM datasets were similarly distributed among all clients.

We observed an improvement of 2% with MSD, 3% with UCSF PDGM, and 2% with UPenn-GBM compared to the results on heterogeneous results (Table 1 of main paper) for 100 communication rounds. Our focus was on demonstrating the effectiveness of our method with data heterogeneity. While results align closely with the homogeneous setup, differences observed are minimal and deemed acceptable for real-time scenarios.

| Dataset | Dice ↑ | IoU ↑ | Prec. ↑ | Sens. ↑ |
|---|---|---|---|---|
| UCSF-PDGM | 0.91 ± 0.02 | 0.78 ± 0.04 | 0.85 ± 0.03 | 0.88 ± 0.03 |
| UPenn-GBM | 0.94 ± 0.01 | 0.74 ± 0.02 | 0.92 ± 0.02 | 0.96 ± 0.01 |
| MSD - Brain | 0.97 ± 0.01 | 0.85 ± 0.03 | 0.96 ± 0.01 | 0.99 ± 0.01 |

Table 8: Comparative results for IID or Homogeneous data, where the same dataset is evenly distributed among all clients.

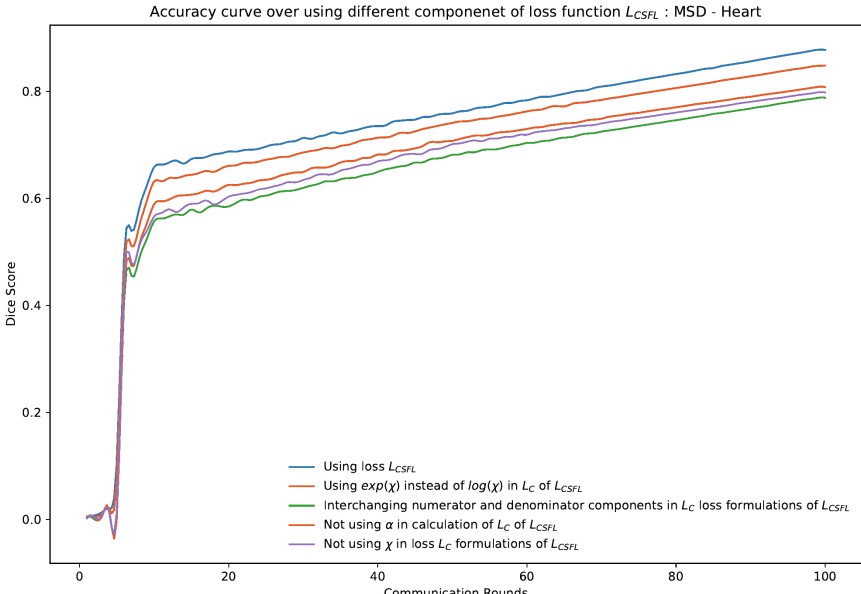

Figure 3: Accuracy plot for varying components of the loss function on MSD - Heart data

## 5.3 SAFCF with Limited Domain Shift

Table 9 displays the test results for individual clients after training within our SAFCF setup for 100 communication rounds, using BraTS 21 Baid et al. (2021) at client 1, BraTS 19 Bakas et al. (2018) at client 2, and the MSD dataset Antonelli et al. (2022) at client 3. The results highlight our SAFCF framework performance across datasets with minimal domain shift. Notably, the MSD dataset is derived from the BraTS challenges, making it a subset within the same domain. The consistent improvement in performance across these datasets indicates that our framework effectively generalizes to variations within the BraTS domain.

| Data | Dice ↑ | IoU ↑ | Prec. ↑ | Sens. ↑ |
|---|---|---|---|---|
| BRATS 21 | $0.90 \pm 0.02$ | $0.81 \pm 0.03$ | $0.87 \pm 0.01$ | $0.92 \pm 0.20$ |
| BRATS 19 | $0.94 \pm 0.02$ | $0.85 \pm 0.04$ | $0.92 \pm 0.03$ | $0.95 \pm 0.03$ |
| MSD - Brain | $0.96 \pm 0.02$ | $0.87 \pm 0.02$ | $0.93 \pm 0.03$ | $0.97 \pm 0.01$ |

Table 9: Ablation results obtained from the limited domain shift datasets provided at each client within SAFCF.

# 6 Conclusion

The non-IID nature of data presents a significant challenge for federated learning effectiveness. In this work, we address this challenge by combining local strategies (CFSL) with a global approach (DSWA) within our SAFCF framework. This framework demonstrated substantial improvements across all datasets, achieving a Dice coefficient increase over baselines by up to 13.6% on the UCSF-PDGM dataset, up to 13.0% on the UPenn-GBM dataset, and as much as 23.2% on the MSD dataset. SAFCF effectively balances spatial localization and model representation by prioritizing clients based on data scales and mitigating model drift due to data heterogeneity through the inclusion of epistemic uncertainty in weighted aggregation. Though SAFCF shows best results, it remains limited in addressing severe data heterogeneity, especially due to the scarcity of diverse healthcare data. Future research will focus on an in-depth examination of SAFCF's performance under significant distribution shifts among local private data, further refining its adaptability in heterogeneous environments.

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

# 7   Appendix

In this appendix, we aim to provide a more comprehensive understanding of our contributions by presenting information, such as implementation details, and more analysis that we could not include in the main content of the paper.

# 8   Implementation Details

## 8.1   Effect of Learning Rate ($lr$)

A grid search across learning rates ranging from 1e-2 to 1e-7 was conducted to identify the optimal value of learning rate based on validate dataset (Table 10). The Dice coefficient used to evaluate segmentation models, was employed to assess performance on the validation set. Results indicates that a learning rate of 1e-5 yielded the highest Dice coefficient (0.86), significantly outperforming other learning rates (range: 0.32-0.55). Based on these findings, the learning rate of 1e-5 was selected for subsequent experiments, optimizing model performance.

Table 10: Dice coefficients for different learning rates on MSD brain data.

| Learning Rate | 1e-2 | 1e-3 | 1e-4 | 1e-5 | 1e-6 | 1e-7 |
|---|---|---|---|---|---|---|
| Dice | 0.32±0.02 | 0.41±0.02 | 0.56±0.02 | **0.86± 0.02** | 0.75±0.02 | 0.55±0.02 |

## 8.2   Effect of $\lambda$

For CFSL, we performed a parameter tuning experiment on the variable $\lambda$, with values ranging from 0 to 1 (0, 0.25, 0.5, 0.75, 1) to achieve optimal results based on validation dataset. Through our experiments, we have observed that setting the $\lambda$ value as 0.5 has yielded favorable outcomes, as it allocates equal priority to both the modified dice and improved contrastive loss. This confirms that spatial precision and feature similarity are equally important for enhancing the model's performance. Table 11 illustrates that changing the value of $\lambda$ affects model accuracy across all datasets. Similarly, for the learning rate $\alpha$, we tested with $e-3$, $e-5$, $e-7$, and observed best optimal value at $e-5$.

Table 11: Effect of SAFCF performance by varying $\lambda$, which balances the modified Dice loss $\mathbb{L}_{\text{dice}}$ and the improved contrastive loss $\mathbb{L}_c$ in CFSL.

| $\lambda$ | UCSF-PDGM | UPenn-GBM | MSD |
|---|---|---|---|
| 0 | 0.69 | 0.72 | 0.76 |
| 0.25 | 0.71 | 0.73 | 0.78 |
| 0.5 | **0.75** | **0.78** | **0.86** |
| 0.75 | 0.72 | 0.74 | 0.76 |
| 1 | 0.72 | 0.71 | 0.75 |

If $\alpha$ exceeds $e-5$, it can lead to rapid, substantial, and unpredictable fluctuations in loss updates, potentially hindering convergence. Conversely, if $\alpha$ is less than $e-5$, the updates may become excessively slow, approaching convergence without actually reaching it, and even achieving convergence may require a significant amount of time.

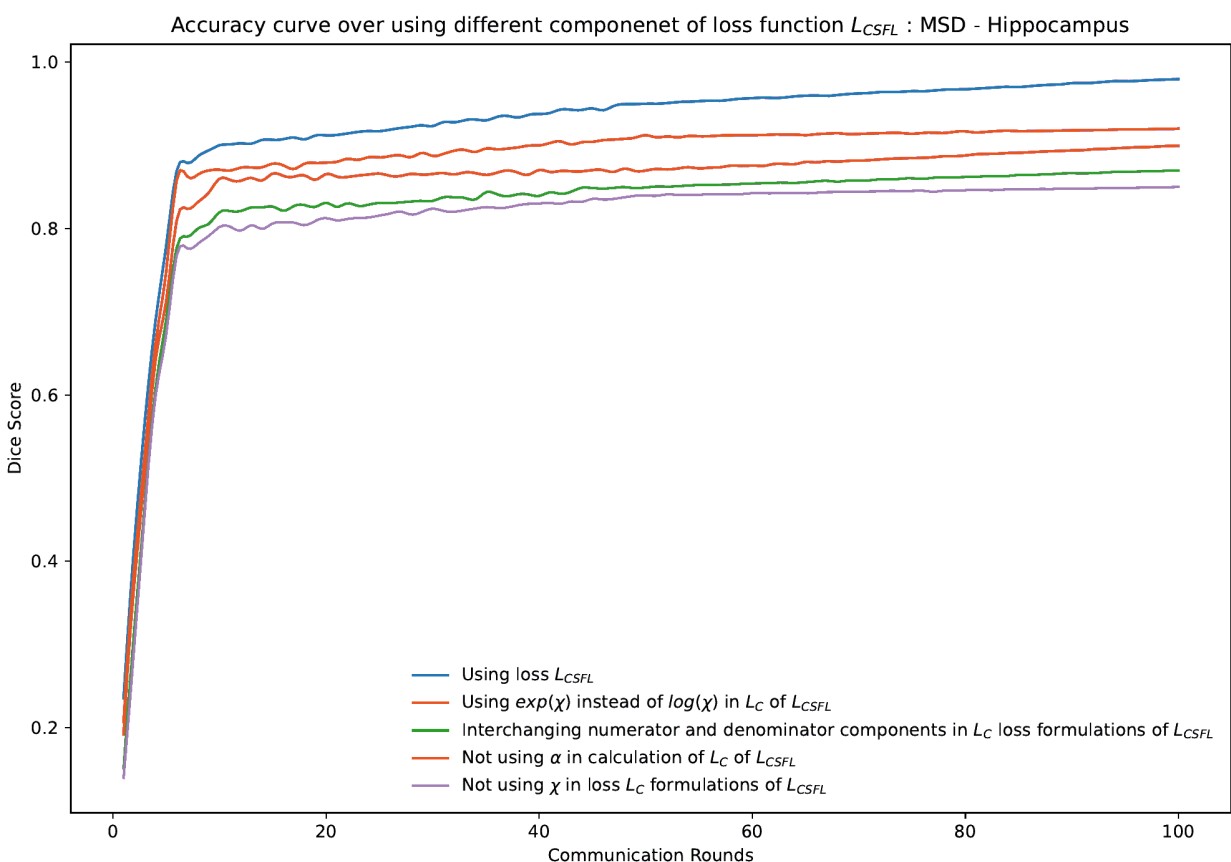

Figure 4: Accuracy plot for varying components of the loss function on MSD - Hippocampus data

## 8.3 Empirical Analysis on $L_c$

We study the effect of different components of $L_c$ on the accuracy of our framework. The results are shown in Figure 4, 5, 6. From the accuracy plots it is observed that all the components such as $\alpha$, $\chi$, $\log \chi$ are equally important in our $L_c$ for better optimization.

## 8.4 Segmentation Models

We assessed several segmentation models as alternatives to the 3D U-Net in our SAFCF framework, including YoloV9e, SAM3D, and Autoencoder, across three datasets: UCSF PDGM, UPenn-GBM, and MSD. As shown in Table 12, the performance of these models was generally lower compared to the 3D U-Net (Table 1 of main paper). Consequently, we chose the 3D U-Net as the preferred model for our framework.

| Data | Network | Dice ↑ | IoU ↑ | Prec. ↑ | Sens. ↑ |
|------|---------|--------|-------|---------|---------|
| UCSF PDGM | YoloV9e | 0.44±0.03 | 0.39±0.01 | 0.41±0.04 | 0.42±0.02 |
| | SAM3D | 0.43±0.01 | 0.35±0.03 | 0.38±0.02 | 0.41±0.05 |
| | Autoencoder | 0.82±0.01 | 0.69±0.03 | 0.78±0.02 | 0.85±0.05 |
| UPenn-GBM | YoloV9e | 0.46±0.02 | 0.41±0.03 | 0.43±0.03 | 0.45±0.02 |
| | SAM3D | 0.45±0.04 | 0.37±0.02 | 0.41±0.02 | 0.44±0.03 |
| | Autoencoder | 0.84±0.01 | 0.71 ±0.03 | 0.81±0.02 | 0.88±0.05 |
| MSD | YoloV9e | 0.48±0.02 | 0.42±0.03 | 0.45±0.03 | 0.47±0.02 |
| | SAM3D | 0.47±0.03 | 0.41±0.02 | 0.42±0.02 | 0.47±0.05 |
| | Autoencoder | 0.86±0.01 | 0.75±0.03 | 0.84±0.02 | 0.87±0.05 |

Table 12: Comparative analysis of different segmentation models within SAFCF.

## 8.5 Communication Efficiency

In our experimental observations, we found that training the local model with our proposed loss takes less time than simCLR and InfoNCE and almost the same time as focal tversky loss. Additionally, simCLR and InfoNCE show a slight advantage in the first ten communication rounds, but our approach surpasses both methods for 50 and 100 communication rounds.

With server aggregation methods, we noticed that as the number of communication rounds increases significantly, the accuracy of all approaches tends to decrease. This decline is attributed to the divergence of local updates, meaning that the local optimal solutions do not align consistently with the global optimum. Despite this challenge, our DSWA method distinctly outperforms other server aggregation approaches, demonstrating its robustness in handling the impact of prolonged local training epochs.

Figure 9 illustrates the average training time per communication round for various aggregation methods and local contrastive loss functions in our SAFCF framework across three different medical imaging datasets: UCSF-PDGM, UPenn-GBM, and MSD. The results reveal differences in computational efficiency among the methods. As observed from the graph, FedProx, Scaffold, FedNova, and FedMA exhibit progressively longer training times, with FedMA taking the longest per round. Although FedAvg requires less computation time compared to our SAFCF, the accuracy achieved by SAFCF within the same time frame is superior.

## 8.6 Qualitative Analysis

The qualitative results of our proposed approach are shown in Figure 8 for non-IID data from (a) MSD - Brain, (b) MSD - Heart, and (c) MSD - Hippocampus, each provided to individual clients within our SAFCF framework. A significant resemblance between the predicted masks and the ground truths highlights the effectiveness of our framework.

**Qualitative Results:** In Figure 11, the qualitative outcomes of our proposed method are compared with alternative loss techniques. It is evident that our approach exhibits a significant likeness between the predicted masks and the ground truths. This resemblance can be attributed to the capacity of our method to

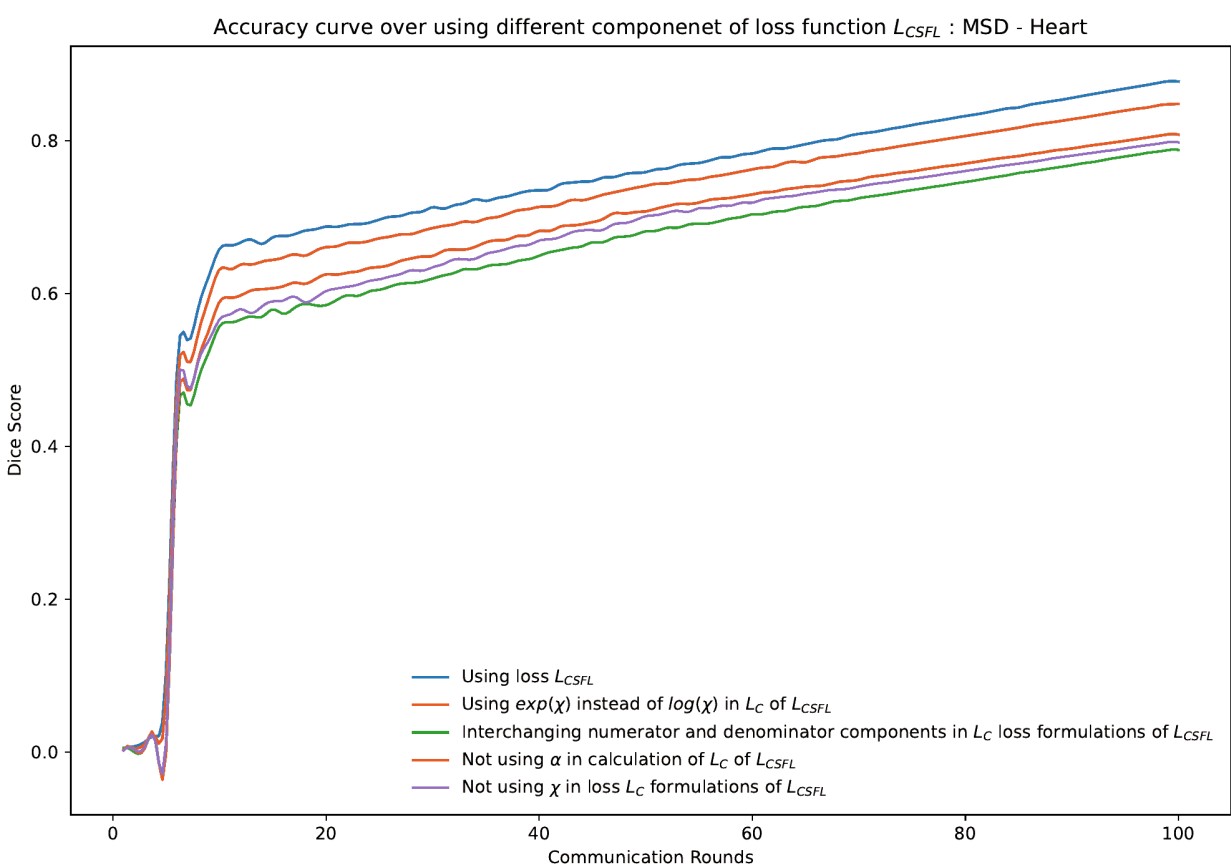

Figure 5: Accuracy plot for varying components of the loss function on MSD - Heart data

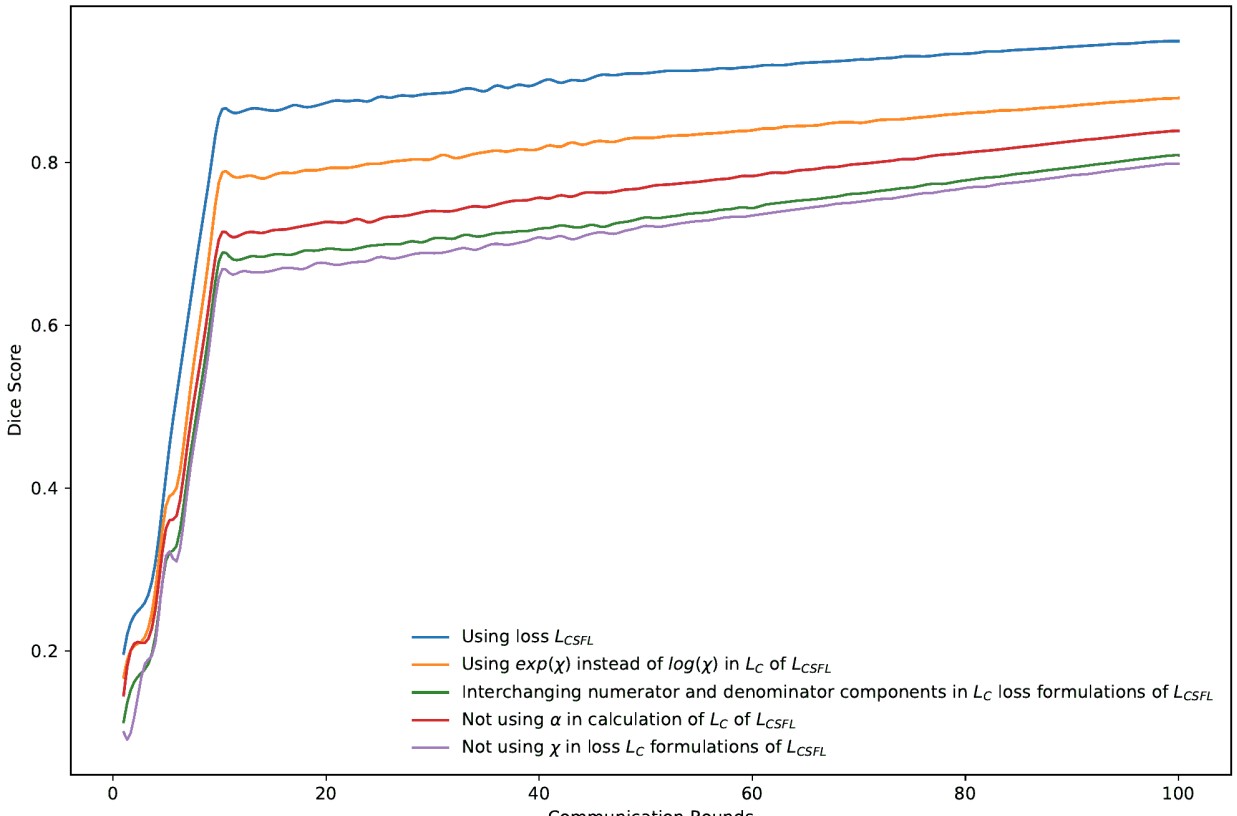

Figure 6: Accuracy plot over using different component of loss function on MSD - Brain data

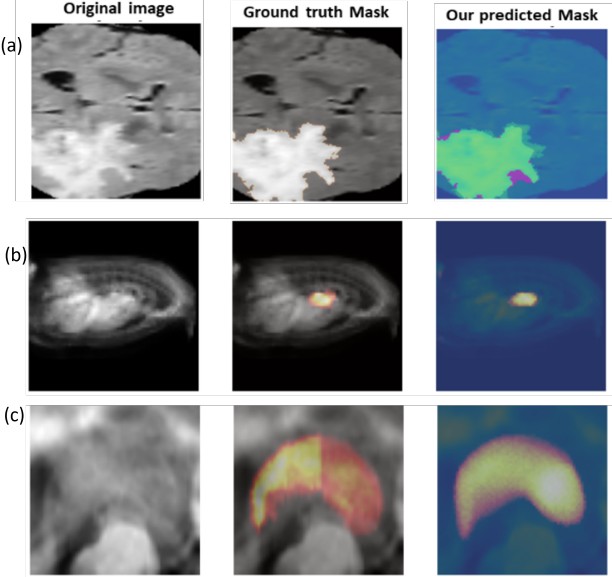

Figure 7: Qualitative results obtained from non-iid data (a) MSD - Brain (b) MSD - Heart (c) MSD - Hippocampus at each individual client of our SAFCF framework

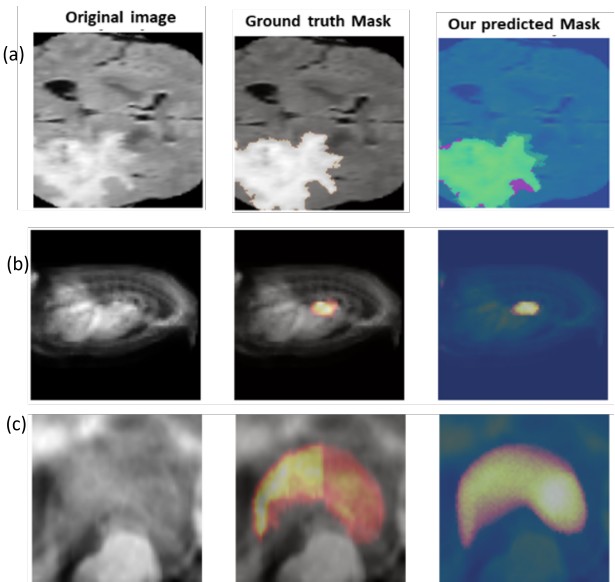

Figure 8: Qualitative results obtained from non-iid data (a) MSD - Brain (b) MSD - Heart (c) MSD - Hippocampus at each individual client of our SAFCF framework

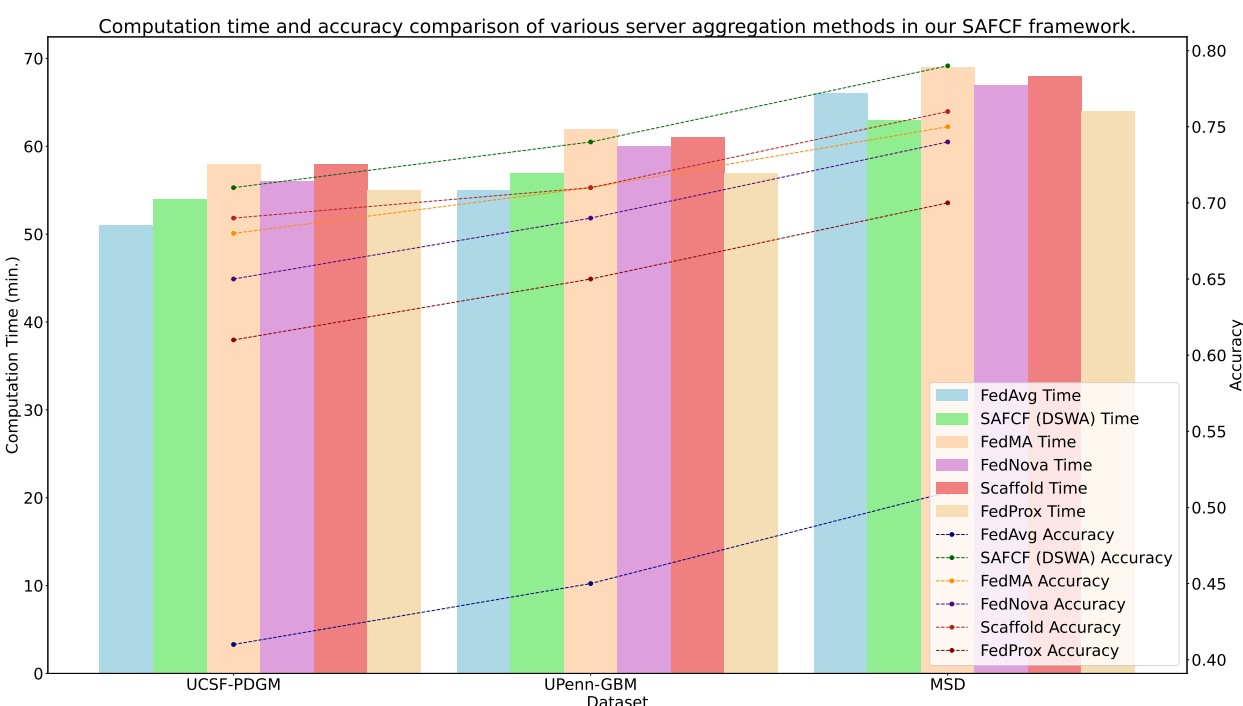

Figure 9: Computation efficiency and accuracy comparison of various aggregation methods in our SAFCF framework.

effectively capture spatial and feature information across all dimensions of an image, distinguishing it from other approaches.

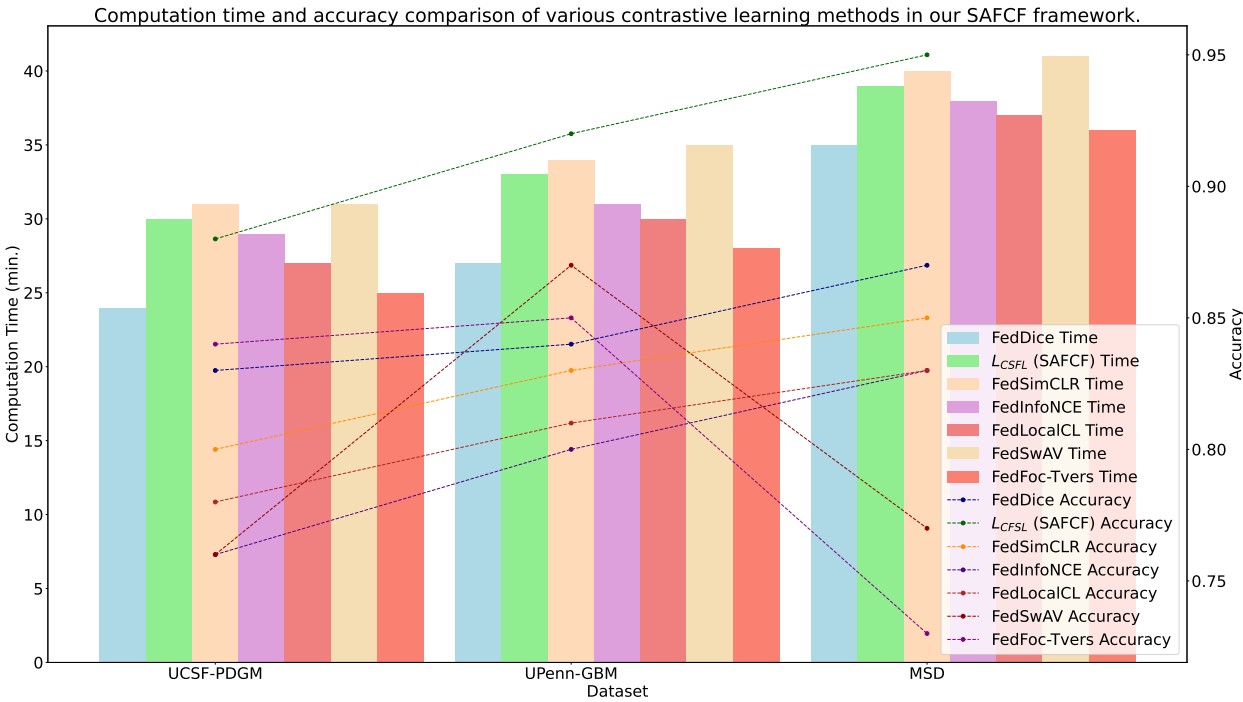

Figure 10: Computation efficiency and accuracy comparison of various local contrastive loss functions in our SAFCF framework.

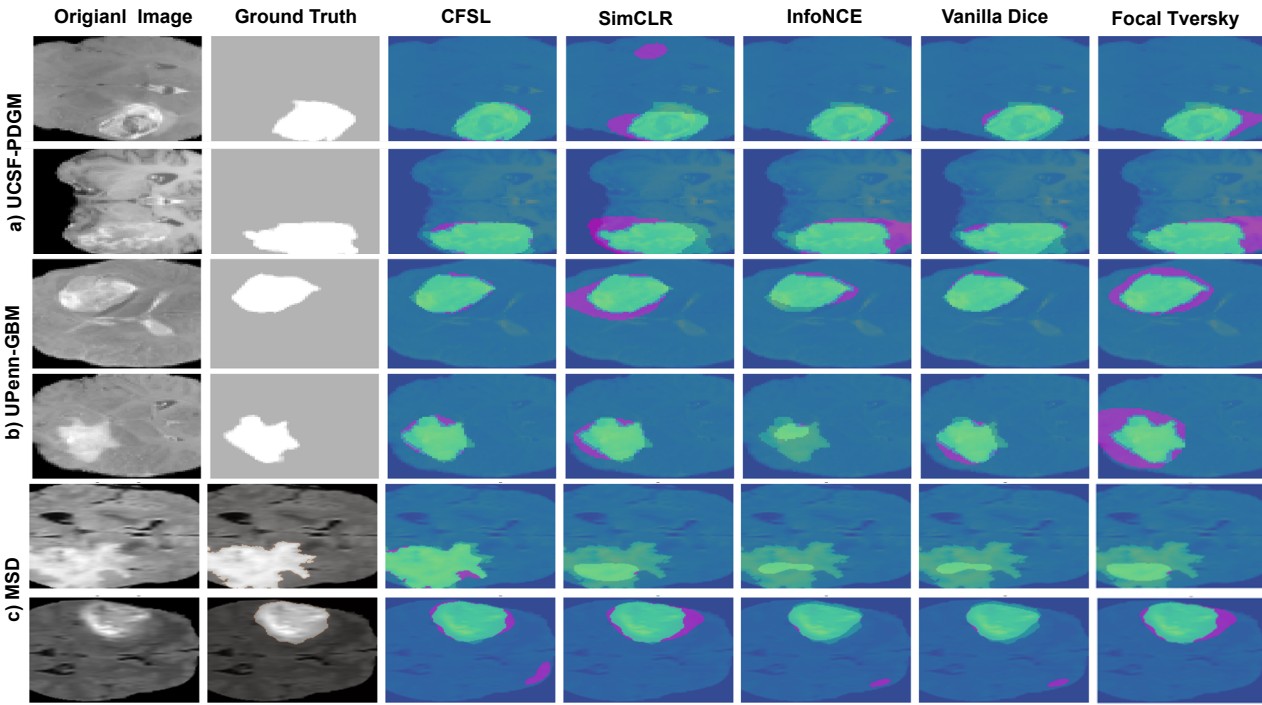

Figure 11: Visualization result with ours and baselines, a) UCSF-PDGM, b) UPenn-GBM, c) MSD.

