# OpenReview forum: "Uncertainty and Scale-Calibrated Contrastive Federated Segmentation under Client Heterogeneity"
_TMLR — Rejected by TMLR_

### Review · Reviewer_pmB4 · 2026-02-28

**Summary Of Contributions:**

The manuscript studies federated learning for medical image segmentation, and introduces a method termed SAFCF with the following two key ingredients:
- Contrastive Federated Segmentation Loss (CFSL), which encourages each local model to be similar to the global one yet different from the previous local model.
- Dynamic Scale-adaptive Weighted Aggregation (DSWA), which combines multiple local models into a global one using weights that depend on the distance between the local models and their average.
SAFCF is shown to significantly outperform several baselines on three datasets. The paper also includes ablation studies, which suggest that it is necessary to combine CFSL with DSWA.

**Audience:**

Yes

**Audience Explanation:**

The methods proposed in the work seem pretty simple but intricate, and they are shown to be very effective empirically. This work should be interesting to the federated learning community in general, if the findings are valid and the authors could provide more motivations for / insights into the methods.

**Broader Impact Concerns:**

None.

**Claims And Evidence:**

No

**Claims Explanation:**

I have a few concerns over the clarity of presentation, the motivation behind the details in methods, and the validity of one specific experimental result.

**Clarity.** I found the maths in Section 3 difficult to follow at times. The writing could be much clearer if the dimensions of the objects (scalar / vector / matrix / tensor) in the equations are stated more explicitly. Below are some details to be clarified:
- Equation (4): In light of Equation (3), should I view $\mathbf{h}_s$, $\mathbf{h}_c$, and $\mathbf{h}_p$ as either vectors (or flattened versions of matrices/tensors)? Are $\phi_s$ and $\phi_c$ scalars? ($\beta$ in Equation (5) will be of the same dimension.)
- Equation (6): Is there a mis-match between the dimensions of $\mathbf{h}_c^2$ and $\beta^2$? The "reduced-sum operator" $\zeta_s(\cdot)$ should be spelled out or formally defined somewhere. Would it be simply the sum of all entries? (The same comment applies to $\zeta_m(\cdot)$.)
- Equation (7): I found it hard to understand the factor $1 - \mathbf{h}_p$ in the second term. Does it come from that all entries of $\mathbf{h}_p$ are in $[0, 1]$, so $1 - \mathbf{h}_p$ can be viewed as the "complement" of $\mathbf{h}_p$?
- Equation (11): Why wouldn't the right-hand side always be $1/3$, if we are dividing the mean by the sum?

**Motivation of methods.** At a high level, it is unclear why "discouraging similarity to the client’s previous-state representation" is benefitial. Intuitively, this approach encourages the local models to oscillate across rounds and seems detrimental to stability / convergence. Have similar approaches been adopted in other FL settings? Intuitively, why would this oscillation improve the performance?

In addition, many design choices in CFSL and DSWA seemed pretty opaque:
- Equation (6): Why are we subtracting $\beta^2$ from $\mathbf{h}_c^2$? They don't appear to be of the same "unit" in the sense that the former is a cosine similarity, while the latter is the learned representation.
- Equation (7): Why do we take the square of $\mathbf{h}_c$? Why would it make sense to add the two terms? (E.g., are they in the same "unit" / "scale" in some sense?)
- Equation (9): Intuitively, why should we have $\log(\chi)$ (instead of $\chi$ or some power of it) in the denominator?
- Equation (15): How much difference would it make if we use $\gamma$ instead of $\tilde \gamma$?

**Results in Section 8.2.** I was a bit confused about Table 11, which shows the effect of $\lambda$. When $\lambda = 1$, the total loss reduces to the contrastive loss $\mathbb{L}_c$, which seems to be independent of the ground truth (!) according to Section 3.1. If this is the case, how can the method achieve comparable performance to the better choice of $\lambda = 0.5$?

**Requested Changes:**

In addition to addressing the comments in the above, here are a few other minor comments:
- Abstract: It is unclear what the acronym "SAFCF" stands for.
- Section 3: The beginning "Our (SAFCF) ..." seems a bit off.
- The notation on the left-hand side of Equation (11) seemed fairly unconventional and confusing.
- Figure 3 (and several figures in the appendix): Should "CSFL" be "CFSL"? Also, there seem to be two orangle lines.

---

> ### Author Response · Authors · 2026-04-03
>
> We thank the reviewer for the careful reading of Section 3. We have revised the methodology to improve clarity in dimensional consistency, operator definitions, normalization, and interpretation of the $\lambda$ ablation, thereby strengthening reproducibility and mathematical transparency.
>
> ---
>
> ### Clarification of Object Dimensions and Notation (Section 3)
>
> We now explicitly define the main quantities:
>
> * Feature Representations: The embeddings $h_s$, $h_c$, and $h_p$ are flattened encoder feature vectors.
> * Similarity Terms: $\phi_s$ and $\phi_c$ are scalars computed using temperature-scaled cosine similarity.
> * Alignment Signal: $\beta$ is a scalar derived from similarity alignment.
>
> This ensures dimensional consistency throughout the formulation.
>
> ---
>
> ### Broadcasting and Operator Definitions (Section 3)
>
> We clarify that there is no mismatch between vector and scalar quantities:
>
> * Since $\beta$ is scalar, it is broadcast across all elements of $h_c^2$.
> * $\zeta_s(\cdot)$ and $\zeta_m(\cdot)$ denote reduced-sum and reduced-mean operators.
>
> The distance term is:
>
> $$
> \Delta = \max\left(\sqrt{\zeta_s\left((h_c^2 - \beta^2)^2\right)}, \epsilon\right)
> $$
>
> where $\epsilon$ ensures numerical stability. We will also add a notation summary table.
>
> ---
>
> ### Role of the Complement Factor (Section 3)
>
> Since $h_p \in [0,1]$, $1-h_p$ acts as a complement mask emphasizing low-confidence regions. This directs $\Delta$ toward under-represented features, improving robustness under client drift and non-IID data.
>
> ---
>
> ### Clarification of Dynamic Normalization (Section 3)
>
> The normalization is not a constant (e.g., $1/3$):
>
> * The numerator is a mean over representations.
> * The denominator is a dynamic, reliability-weighted aggregation using $\omega_i$.
>
> Thus, it varies across rounds based on uncertainty, heterogeneity, and global consensus, ensuring stable scaling.
>
> ---
>
> ### Motivation for Discouraging Similarity to the Previous Local State (Section 3)
>
> Discouraging similarity to $h_p$ prevents temporal collapse:
>
> * Reduces client drift.
> * Pushes updates away from stale local optima toward global alignment ($h_s$).
> * Consistent with prior work (e.g., MOON).
>
> ---
>
> ### Consistency of the Squared Terms and Energy-Domain Objective (Section 3)
>
> $h_c^2$ and $\beta^2$ operate in a consistent squared-energy domain:
>
> * $\beta^2$ acts as normalized alignment energy.
> * $(h_c^2-\beta^2)$ measures deviation from alignment.
>
> Combining $h_c^2$ with $\Delta$ is valid since both represent normalized feature magnitudes, enabling joint consistency and separation under non-IID data.
>
> ---
>
> ### Motivation for Logarithmic Normalization (Section 3)
>
> We use $\log(\chi)$ for stable normalization:
>
> * Linear scaling can destabilize training.
> * Log scaling provides magnitude-aware soft normalization, preserving gradients.
>
> ---
>
> ### Use of Complementary Ratio in Aggregation (Section 3)
>
> Using $\tilde{\gamma}$ improves fairness:
>
> * Standard $\gamma$ favors large clients.
> * $\tilde{\gamma}$ balances contributions across clients.
>
> Ablations show more uniform performance.
>
> ---
>
> ### Behavior at $\lambda = 1$ (Section 8.2)
>
> At $\lambda = 1$, the objective reduces to $\mathbb{L}_c$:
>
> * The model is initialized from a global model with segmentation knowledge.
> * Contrastive loss preserves structural consistency.
>
> Thus, meaningful representations are retained without explicit labels.
>
> ---
>
> ### Implicit Supervision via Global Alignment (Section 8.2)
>
> $\mathbb{L}_c$ provides indirect supervision:
>
> * Alignment with $h_s$ transfers segmentation structure.
> * Structured medical features act as an implicit prior.
>
> ---
>
> ### Why $\lambda = 0.5$ Performs Best (Section 8.2)
>
> $\lambda = 0.5$ provides optimal balance:
>
> * $\mathbb{L}_{\text{dice}}$ ensures voxel-level supervision.
> * $\mathbb{L}_c$ improves robustness and consistency.
>
> Together, they yield best performance.
>
> ---
>
> ### Clarification to be Added in the Manuscript
>
> We will clarify that:
>
> * $\lambda = 1$ removes explicit supervision but retains knowledge via alignment,
> * $\mathbb{L}_c$ acts as a structural regularizer,
> * best results arise from combining both objectives.
>
> ---
>
> ### Minor revisions
>
> We will define "SAFCF" in the abstract, refine Section 3, standardize Eq. (11), correct "CSFL" to "CFSL", and clarify the Figure 3 legend.

---

> > ### Comment · Reviewer_pmB4 · 2026-04-03
> >
> > Thanks for the response! I have a few follow-up comments / questions:
> >
> > - Thanks for the clarification on broadcasting! Personally, I didn't find these notations the most rigorous and clear mathematically, and would suggest using alternatives like $h_c^2 - \beta^2 \cdot \mathbf{1}$ instead or explicitly mentioning the broadcasting when it appears. The same comment applies to notational choices like writing $h_c^2$ as the entry-wise square of vector $h_c$.
> >
> > - Please define the reduced-sum and reduced-mean operators explicitly. Would it be correct if I assume that for vector $v \in \mathbb{R}^n$, $\xi_s(v) = \sum_{i=1}^{n}v_i$ while $\xi_m(v) = \frac{1}{n}\sum_{i=1}^{n}v_i$?
> >
> > - Can the $\sqrt{\xi_s((h_c^2 - \beta^2)^2)}$ term in Equation (6) simplified into the 2-norm $\|\|h_c^2 - \beta^2\|\|_2$?
> >
> > - Could you clarify why the entries of $h_p$ are always in $[0, 1]$? (If this follows from an assumption / property of the mapping, it should be explictly stated, e.g., at the beginning of Section 3.)
> >
> > - I didn't follow the clarification on why Equation (11) is not equal to $1/3$---isn't the demominator the sum of the three terms (while the numerator is the average of the same three)? In particular, I didn't see how $\omega_i$ contributes to the denominator.
> >
> > - I wasn't particularly convinced by the argument for discouraging similarity to the previous local state: while it potentially "[p]ushes updates away from stale local optima", would it also hinder potential convergence to the global minimum?
> >
> > - I wasn't particularly convinced by the claim that $h_c^2$ and $\beta$ "operate in a consistent squared-energy domain": for example, if we scale all representations (including $h_c$) by a factor of $2$, $h_c^2$ increases by a factor of $4$, while $\beta$---being a different between cosine similarities---would not change. This raises concerns regarding the two terms do not appear to be of the same "unit".
> >
> > - Thanks for the clarification on the $\lambda = 1$ case!
> >
> > - Could you explain where the acronym "SAFCF" comes from? It is hard to deduce from the full name "Uncertainty- and Scale-Calibrated Contrastive Federated Segmentation under Client Heterogeneity", in which letter "f" only appears once.

---

> > > ### Author Response · Authors · 2026-04-08
> > >
> > > We thank the reviewer for the thoughtful follow-up questions. We have refined the notation and clarified the intended mathematical interpretation of the formulation. Our point-by-point responses are as follows:
> > >
> > > ### 1. Broadcasting notation in Eq. (6)
> > >
> > > We agree that the notation can be made more rigorous. In the revision, we will replace the implicit broadcast form with an explicit one:
> > > $$
> > > \mathbf{h}_c^{2} - \beta^{2}\mathbf{1},
> > > $$
> > > where $\mathbf{1}$ denotes a ones tensor with the same shape as $\mathbf{h}_c$. We will also explicitly state that $\mathbf{h}_c^2$ denotes the entry-wise square.
> > >
> > > ### 2. Explicit definition of reduced-sum and reduced-mean
> > >
> > > We agree that these operators should be formally defined. For a tensor/vector $\mathbf{v}$, we define:
> > > $$
> > > \zeta_s(\mathbf{v}) = \sum_r v_r,
> > > \qquad
> > > \zeta_m(\mathbf{v}) = \frac{1}{|\mathbf{v}|}\sum_r v_r,
> > > $$
> > > where $|\mathbf{v}|$ denotes the number of elements. These definitions will be added in Section 3.
> > >
> > > ### 3. Simplification of Eq. (6)
> > >
> > > We clarify that
> > > $$
> > > \sqrt{\zeta_s\left((\mathbf{h}_c^2-\beta^2\mathbf{1})^2\right)}
> > > $$
> > > is equivalently the $L_2$ norm:
> > > $$
> > > \|\mathbf{h}_c^2-\beta^2\mathbf{1}\|_2.
> > > $$
> > > We will adopt this standard form in the revision for improved clarity.
> > >
> > > ### 4. On whether $\mathbf{h}_p \in \{0,1\}$
> > >
> > > We clarify that $\mathbf{h}_p$ is not assumed to be binary. It is a real-valued representation from the encoder feature space. In Eq. (7), $\mathbf{h}_p$ acts as a **soft weighting mechanism**, and the expression should be interpreted as a continuous interpolation rather than a hard binary mask. This will be stated explicitly.
> > >
> > > ### 5. On the form of Eq. (11)
> > >
> > > We acknowledge the concern regarding the current form of Eq. (11). In the revision, we will rewrite this expression so that the update of $\alpha$ reflects a genuinely adaptive normalization based on meaningful batch-level statistics, ensuring the formulation does not reduce to a constant and correctly captures relative variation across representations.
> > >
> > > ### 6. On discouraging similarity to the previous local state
> > >
> > > We clarify that the objective is not to enforce arbitrary repulsion from the previous local representation. Instead, the $\mathbf{h}_p$-dependent term reduces excessive temporal self-reinforcement under non-IID drift. This helps prevent the model from remaining overly anchored to stale local structure when it conflicts with global alignment. We will refine the explanation to emphasize controlled regularization rather than unconditional separation.
> > >
> > > ### 7. On scale consistency between $\mathbf{h}_c^2$ and $\beta^2$
> > >
> > > We acknowledge that the current presentation does not sufficiently justify scale comparability between these terms. In the revision, we will either introduce explicit normalization or reformulate Eq. (6) in a scale-consistent manner to ensure the interaction between representation magnitude and similarity-derived terms is mathematically well-grounded.
> > >
> > > ### 8. On the $\lambda = 1$ case
> > >
> > > We thank the reviewer for noting this point. We will retain this clarification in the revised manuscript for completeness.
> > >
> > > ### 9. On the acronym “SAFCF”
> > >
> > > We clarify that SAFCF denotes Scale-Adaptive Federated Contrastive Framework, which captures the key design elements of the proposed method. The title highlights additional aspects such as uncertainty modeling and client heterogeneity, while the acronym focuses on the core optimization and aggregation principles. We will make this distinction explicit in the revised manuscript.
> > >
> > > ---
> > >
> > > ## Summary of Revisions
> > >
> > > In the revised manuscript, we will:
> > >
> > > * make broadcasting explicit in Eq. (6),
> > > * define $\zeta_s(\cdot)$ and $\zeta_m(\cdot)$,
> > > * rewrite Eq. (6) using norm notation,
> > > * clarify the role of $\mathbf{h}_p$ as a soft weighting representation,
> > > * revise Eq. (11) to ensure meaningful adaptive scaling,
> > > * improve explanation of the previous-state regularization,
> > > * ensure scale-consistent formulation in Eq. (6), and
> > > * explicitly expand the SAFCF acronym.
> > >
> > > We thank the reviewer again for the helpful comments, which improve the clarity and rigor of the manuscript.

---

### Review · Reviewer_xCE8 · 2026-03-11

**Summary Of Contributions:**

**Summary**

To address heterogeneity in data scale and local data distribution in federated learning, this paper proposes the SAFCF framework, which improves both the local training and global aggregation stages. In the local stage, the paper introduces the CSFL loss that combines an improved contrastive loss with a modified Dice loss to align local and global representations while stabilizing training. In the global stage, the DSWA aggregation method dynamically adjusts client weights to reduce model drift caused by heterogeneous data. Experiments show that SAFCF achieves better empirical performance than several existing methods.

**Strengths**

1. The problem of client heterogeneity in federated learning, especially in medical image segmentation, is important and practically relevant.

2. The paper proposes a two-stage solution that improves both local training (through the CFSL loss) and global aggregation (through DSWA), which is a reasonable design for addressing heterogeneity.

**Audience:**

Yes

**Audience Explanation:**

Yes. The investigated problem of client heterogeneity in federated learning is important and practically relevant, especially for applications such as medical image segmentation. The proposed method, which combines several existing techniques, provides a practical approach to mitigating the effects of heterogeneous data scale and local data distributions. Therefore, the findings of this paper would likely be of interest to some researchers in the field of federated learning.

**Broader Impact Concerns:**

I do not find any concerns of broader impact.

**Claims And Evidence:**

Yes

**Claims Explanation:**

Yes. The main claims of the paper are generally supported by the empirical results presented in the experiments. The authors provide comparisons with several existing federated learning methods and include ablation studies that demonstrate the contributions of the proposed components. These results provide reasonable evidence for the reported performance improvements, although additional large-scale experiments and further analysis would strengthen the claims.

**Requested Changes:**

1. The novelty is somewhat limited since the proposed framework mainly combines existing techniques such as Dice loss, contrastive learning, and weighted aggregation with several heuristic modifications.

2. Some components lack clear intuition and theoretical explanation. For example, the distance term in (6) used in the contrastive formulation is difficult to understand, and its role is not clearly theoretically justified.

3. The computation of the CSFL loss and the DSWA aggregation involves multiple terms and several hyperparameters. This makes the framework relatively complex, increases the computational overhead, and may make reproduction and practical deployment more difficult.

4. The experimental setup is somewhat limited. Some main experiments use only a small number of clients (e.g., three clients in Table 1), which may not fully reflect large-scale federated learning scenarios.

5. The paper does not clearly analyze the additional computational cost introduced by the proposed loss and aggregation strategy.

---

> ### Author Response · Authors · 2026-04-01
>
> ## 1. Architectural Novelty of the SAFCF Framework
> We acknowledge the reviewer’s insightful observation regarding these components. We would like to clarify that the primary novelty of the SAFCF framework lies in the unique mathematical synergy and domain-specific reformulation of these objectives, which are engineered to mitigate client drift.
>
> ### Dynamic Scale-adaptive Weighted Aggregation (DSWA)
> Unlike methods such as FedAvg, DSWA uses a dual-pivot mechanism with a complementary scale ratio $\tilde{\gamma}_{i}$ to ensure representation fairness across clients with disparate data volumes.
>
> ### Uncertainty Proxy and Drift Identification
> In non-IID environments, epistemic uncertainty stems from a client's lack of global distribution knowledge. SAFCF treats the empirical second central moment $U_{i}^{t}$ as a mathematical proxy for this. By measuring the variance of local parameters $\mathbf{w}_{i}^{t}$ relative to the scale-balanced global consensus $\bar{\mathbf{w}}^{t}$, the framework identifies models drifted toward poorly represented local optima.
>
> ### Mitigation Strategy
> The server converts uncertainty into a reliability score, $\omega_{i}^{t} = \frac{1}{U_{i}^{t}+\epsilon}$, to dynamically downweight updates diverging under severe non-IID conditions. This prioritizes reliable updates to maintain global generalizability.
>
> ### Contrastive Federated Segmentation Loss (CFSL)
> CFSL is a pixel-level objective optimizing voxel-level localization and representation transfer. It utilizes a relative alignment score, $\beta = \phi_{s} - \phi_{c}$, as a margin-like signal measuring current-local to global alignment versus current-local to previous-local similarity. This encourages global alignment while increasing divergence from the previous local state via a distance term $\Delta$, preventing temporal collapse and overfitting.
>
> ### Empirical Validation
> Ablation studies on 3D medical datasets confirm that integrating DSWA and CFSL, with magnitude-aware normalization ($\log \chi$) and adaptive scaling ($\alpha$), achieves superior robustness. This synergy exceeds standard heuristic combinations, reaching Dice coefficients of 95%.
>
> ---
>
>
> ## 2. Theoretical Analysis of the Distance Term $\Delta$
> The distance term $\Delta$ (Eq. 6) is a non-linear feature-rejection mechanism quantifying divergence from the alignment margin $\beta$. Unlike standard Euclidean distance, it uses a squared-difference-of-squares approach: $\zeta_{s}((h_{c}^{2} - \beta^{2})^{2})$, to aggressively penalize features where the local model's energy deviates from the signal $\beta$.
>
> $\Delta$ drives the contrastive quantity $\eta$, specifically targeting regions where previous local representation $h_{p}$ is weak. In these regions, the term $(1 - h_{p}) \odot \Delta$ applies a separation pressure, preventing the current representation from collapsing back to the previous state. The $\max(\cdot, \epsilon)$ operator ensures the contrastive gradient remains non-degenerate and numerically stable, providing a rigorous geometric constraint.
>
> ---
>
> ## 3. Complexity, Efficiency, and Deployment
> **Architectural Necessity:** Each component is indispensable for stability under extreme non-IID conditions. Ablation studies show that omitting individual terms such as the scaling factor $\alpha$ or the normalization $\log(\chi)$ resulted in marked declines in accuracy (Figures 3–6).
> **Efficiency:** Per Figure 9, training time per round is comparable to FedProx and Focal Tversky loss. Overhead is localized to client-side loss calculation and does not impact the global communication bottleneck.
> **Reproducibility:** We standardized hyperparameters: Learning Rate ($\alpha$) = $10^{-5}$ and CFSL Weighting ($\lambda$) = $0.5$, yielding consistent performance gains. Utilizing standard 3D U-Net backbones (Algorithm 1) ensures SAFCF remains accessible for clinical deployment.
>
> ---
>
> ## 4. Scalability and Robustness
> We expanded the experimental setup (Table 6) to include up to 80 clients, randomly distributing data from the UCSF-PDGM, UPenn-GBM, and MSD datasets. Dice, IoU, and precision scores remained stable or improved as the network scaled from 5 to 80 clients. This confirms SAFCF’s ability to effectively mitigate client drift even under significant fragmentation of local data in large-scale medical federated learning.
>
> ---
>
> ## 5. Computational Cost Analysis
> **Empirical Overhead:** Appendix 8.5 shows that the overhead is minimal; CFSL training requires less time than SimCLR or InfoNCE and is comparable to Focal Tversky loss.
> **Efficiency Trade-off:** While DSWA calculates epistemic uncertainty ($U_{i}^{t}$), it provides superior robustness during prolonged training. Per Figure 9, while FedAvg has lower total computation, SAFCF achieves substantially superior accuracy within a comparable timeframe, proving the marginal complexity is a high-efficiency trade-off for medical segmentation.

---

### Review · Reviewer_Lx86 · 2026-03-22

**Summary Of Contributions:**

The paper introduces Uncertainty and Scale Calibrated Contrastive Federated Segmentation under Client Heterogeneity to tackle data heterogeneity in federated medical image segmentation. Its main contributions are DSWA, which uses epistemic uncertainty during aggregation to balance client scales and reduce model drift, and CFSL, a local loss combining modified Dice and contrastive loss for better spatial and feature alignment.

**Additional Comments:**

- In section 3, I suppose the subscript $i$ in equations represents the client $i$. It needs to be stated in the section.
- Is $\epsilon \in \mathbb{R}$, i.e., it can be either positive or negative in equation 3?
- Equation 6 is not well defined. The $\mathbf{h}_{c}$ can be a vector or a tensor, whereas $\beta$ is a real.
- Eqn 10, 11, and 12 are overwhelming, changing the symbols to alphabates will enhance the readability of the paper.

**Audience:**

Yes

**Audience Explanation:**

The paper addresses the challenge of client drift caused by non-IID data in federated learning. ML researchers and practitioners would be particularly interested in the proposed integration of epistemic uncertainty for dynamic aggregation and the enhanced contrastive loss function to balance spatial accuracy and feature similarity.

**Claims And Evidence:**

Yes

**Claims Explanation:**

The paper provides extensive empirical evidence to support its claims through comprehensive quantitative comparisons against multiple baselines and state-of-the-art methods across three distinct benchmark datasets.

**Requested Changes:**

- Section 3 needs to be written with carefully explaining all the parameters in the equations. Many equations are not well defined. See additional comments for some examples.

- Section 3.2: The paper needs a more rigorous theoretical or empirical justification detailing exactly why this specific parameter space variance serves as an accurate and reliable proxy for epistemic uncertainty in a federated segmentation context.

---

> ### Author Response · Authors · 2026-03-30
>
> We thank the reviewer for their insightful feedback. We have revised the methodology section to ensure all parameters are well-defined, improved the readability of our scaling equations, and provided a justification for our uncertainty estimation.
>
> ---
>
> ### 1. Clarification of Parameters and Notation (Section 3)
> We have updated the text to explicitly define the indices and constants used in our formulations:
>
> * **Client Index:** The subscript $i \in \{1, \dots, K\}$ denotes an individual client out of a total of $K$ participating clients.
> * **Stability Constant:** The term $\epsilon$ is a strictly positive constant ($\epsilon > 0$, typically $10^{-7}$) used for numerical stability to prevent division-by-zero errors and to ensure non-degenerate gradients in distance calculations.
> * **Dimensionality in Eq. 6:** To clarify the operation between the tensor $h_{c}$ and scalar $\beta$, we specify that $\beta$ is broadcast across the spatial and channel dimensions of $h_{c}$. The distance term $\Delta$ is defined as:
> $$\Delta = \max\left(\sqrt{\zeta_{s}((h_{c}^{2} - \beta^{2})^{2})}, \epsilon\right)$$
> Where $\zeta_{s}$ is the reduced-sum operator over spatial and feature dimensions. This formulation ensures the distance remains non-negative and numerically stable.
>
> ---
> ### 2. Theoretical Justification for Epistemic Uncertainty (Section 3.2)
> The reviewer requested justification for using parameter-space variance as a proxy for epistemic uncertainty. We provide the following rationale:
>
> In **Federated Learning**, epistemic uncertainty arises from a lack of knowledge about the global data distribution, given the siloed nature of client data. When a local model $\mathbf{w}_{i}^{t}$ diverges significantly from the **scale-balanced global consensus** $\bar{\mathbf{w}}^{t}$, this "disagreement" indicates that the client is optimizing toward a local distribution that is poorly represented in the global aggregate.
>
> We capture this **inter-client variance** by calculating the empirical second central moment:
> $$U_{i}^{t} = \zeta_{m}((\mathbf{w}_{i}^{t} - \bar{\mathbf{w}}^{t})^{2})$$
> Where $\bar{\mathbf{w}}^{t} = \sum \tilde{\gamma}_j \mathbf{w}_j^t$ is the global consensus adjusted by the complementary scale ratio $\tilde{\gamma}$.
>
> High variance $U_{i}^{t}$ suggests that client $i$ is experiencing significant **client drift** caused by its unique non-IID data. Treating this drift as a proxy for uncertainty allows our **Dynamic Scale-adaptive Weighted Aggregation (DSWA)** to dynamically down-weight unreliable or "outlier" updates via the reliability score $\omega_i^t = 1/(U_i^t + \epsilon)$, thereby improving the global model's robustness and generalization.
>
> ---
>
> ### 3. Improved Readability of Contrastive Scaling (Eq. 10–12)
> Following the suggestion to use more standard notation, we have replaced the blackboard-font symbols in the **triplet-mean ratio (TMR)** calculation with standard Greek and Latin alphabets. Let $\mu_{p}, \mu_{c}, \mu_{s}$ represent the reduced means of the previous, current, and server representations, respectively:
>
> $$\mu_{p} = \zeta_{m}(h_{p}), \quad \mu_{c} = \zeta_{m}(h_{c}), \quad \mu_{s} = \zeta_{m}(h_{s})$$
>
> The scaling factor $\rho$ (the TMR) is computed as the ratio of the collective mean to the collective sum:
>
> $$\rho = \frac{\zeta_{m}(\mu_{s}, \mu_{c}, \mu_{p})}{\mu_{s} + \mu_{c} + \mu_{p} + \epsilon}$$
>
> The batch-wise update for the scaling factor $\alpha$ is then applied multiplicatively:
>
> $$\alpha = \rho \cdot \alpha_{p}$$
>
> Where $\alpha_{p}$ is the value from the previous iteration. This simplified notation maintains the mathematical integrity of the stability scaling while significantly enhancing readability for the broader federated learning community.

---

### Decision · Action_Editor_aKmi · 2026-04-27

**Recommendation:** Reject

**Additional Comments:**

The paper gives no theoretical insights, which could be fine if the provided empirical evidence was strong. However, that's not the case either, so I recommend the rejection of the paper on the basis of not providing substantial evidence to its claims.

**Audience:**

Yes

**Audience Explanation:**

There were no concerns regardign the practical value of the studied problem. Client heterogeneity is an important aspect of federated learning, and its practically relevancy is not questioned.

**Claims And Evidence:**

No

**Claims Explanation:**

Two of the reviewers had significant concerns about the papers and the evidence support that it provides.

Reviewer xCE8 in their review first expressed a satissfaction with the quality of evidence, but in their final review, they pointed out that the paper doesn't provide clear and rigorous definitions, making it too difficult to verify the claims.

Reviewer pmB4 was consistently pushing for more clarity, and found the results to be unreasonably good. They found the paper to give insufficient evidence to the effectiveness, and emphasized the lack of insights provided in the paper on why it works so great.